# TOWARDS DOMAIN ADAPTIVE NEURAL CONTEXTUAL BANDITS

**Ziyan Wang[1], Xiaoming Huo[1], Hao Wang[2]**
Georgia Institute of Technology[1]   Rutgers University[2]
`{wzy,huo}@gatech.edu`[1] `hw488@cs.rutgers.edu`[2]

## ABSTRACT

Contextual bandit algorithms are essential for solving real-world decision making problems. In practice, collecting a contextual bandit's feedback from different domains may involve different costs. For example, measuring drug reaction from mice (as a source domain) and humans (as a target domain). Unfortunately, adapting a contextual bandit algorithm from a source domain to a target domain with distribution shift still remains a major challenge and largely unexplored. In this paper, we introduce the first general domain adaptation method for contextual bandits. Our approach learns a bandit model for the target domain by collecting feedback from the source domain. Our theoretical analysis shows that our algorithm maintains a sub-linear regret bound even adapting across domains. Empirical results show that our approach outperforms the state-of-the-art contextual bandit algorithms on real-world datasets. Code will soon be available at `https://github.com/Wang-ML-Lab/DABand`.

## 1 INTRODUCTION

Contextual bandit (CB) algorithms have shown great promise for naturally handling exploration/exploitation trade-off problems with optimal statistical properties. Notably, LinUCB (Li et al., 2010) and its various adaptations (Yue & Guestrin, 2011; Agarwal et al., 2014; Li et al., 2016; Kveton et al., 2017; Foster et al., 2018; Korda et al., 2016; Mahadik et al., 2020; Zhou et al., 2019), have been shown to be able learn the optimal strategy when all data come from the same domain. However, these methods fall short when applied to data from a new domain. For example, a drug reaction prediction model trained by collecting feedback from mice (the source domain) may not work for humans (the target domain).

So, how does one effectively explore a high-cost target domain by only collecting feedback from a low-cost source domain, e.g., exploring drug reaction in humans by collecting feedback from mice or exploring real-world environments by collecting feedback from simulated environments? The challenge of effective cross-domain exploration is multifaceted: (1) The need for *effective exploration* in both the source and target domains depends on *the quality of representations* learned in the source domain, which in turn requires *effective exploration*; this leads to a chicken-and-egg dilemma. (2) Aligning source-domain representations with target-domain representations is nontrivial in bandit settings, where ground-truth may still be unknown if the action is incorrect. (3) Balancing these aspects – effective exploration and accurate alignment in bandit settings – is also nontrivial.

To address these challenges, as the first step, we allow our method to simultaneously perform *effective exploration* and *representation alignment*, leveraging unlabeled data from both the source and target domains. Interestingly, our theoretical analysis reveals that naively doing so leads to sub-optimal accuracy/regret (verified by our empirical results) and naturally leads to additional terms in the target-domain regret bound. We then follow the regret bound derived from our analysis to develop an algorithm that adaptively collect feedback from the source domain while aligning representations from the source and target domains. Our contributions are outlined as follows:

- We identify the problem of contextual bandits across domains and propose domain-adaptive contextual bandits (DABand) as a the first general method to explore a high-cost target domain while only collecting feedback from a low-cost source domain.

- Our theoretical analysis shows that our method can achieve a sub-linear regret bound in the target domain.
- Our empirical results on real-world datasets show our DABand significantly improve performance over the state-of-the-art contextual bandit methods when adapting across domains.

## 2 RELATED WORK

**Contextual Bandits.** In the realm of adaptive decision-making, contextual bandit algorithms, epitomized by LinUCB (Li et al., 2010), have carved a niche in efficiently balancing the exploitation-exploration paradigm leveraging estimation of reward and uncertainty (Wang et al., 2022; 2016); they have been influential across a spectrum of use cases, notably in complex adaptive systems such as recommender systems (Li et al., 2010; Wang et al., 2022). These algorithmic frameworks, along with their myriad adaptations (Yue & Guestrin, 2011; Agarwal et al., 2014; Li et al., 2016; Korda et al., 2016; Kveton et al., 2017; Foster et al., 2018; Zhou et al., 2019; Mahadik et al., 2020; Xu et al., 2020), have decisively outperformed conventional bandit models that operate devoid of contextual awareness (Auer, 2002). This superiority is underpinned by theoretical models, paralleling the insights in (Auer, 2002), where LinUCB variants are validated to conform to optimal regret boundaries in targeted scenarios (Chu et al., 2011). However, a discernible gap in these methodologies is their reduced efficacy in a new domain, particularly when getting feedback involves high cost. Our proposed DABand bridges this gap by adeptly aligning both source-domain and target-domain representations in the latent space, thereby reducing performance drop when transfer across different domains.

**Domain Adaptation.** The landscape of domain adaptation has been extensively explored, as evidenced by a breadth of research (Pan & Yang, 2009; Pan et al., 2010; Long et al., 2018; Saito et al., 2018; Sankaranarayanan et al., 2018; Zhang et al., 2019; Peng et al., 2019; Chen et al., 2019; Dai et al., 2019; Wang et al., 2020; Nguyen-Meidine et al., 2021; Xu et al., 2023). The primary objective of these studies has been the alignment of source and target domain distributions to facilitate the effective generalization of models trained on labeled source data to unlabeled target data. This alignment is conventionally attained either through the direct matching of distributional statistics (Pan et al., 2010; Tzeng et al., 2014; Sun & Saenko, 2016; Peng et al., 2019; Nguyen-Meidine et al., 2021) or via the integration of an adversarial loss (Ganin et al., 2016b; Zhao et al., 2017; Tzeng et al., 2017; Zhang et al., 2019; Kuroki et al., 2019; Chen et al., 2019; Dai et al., 2019; Wang et al., 2020; Xu et al., 2022; Liu et al., 2023; Shi & Wang, 2023). The latter, known as adversarial domain adaptation, has surged in prominence, bolstered by its theoretical foundation (Goodfellow et al., 2014; Zhao et al., 2018; Zhang et al., 2019; Zhao et al., 2019), the adaptability of end-to-end training in neural network architectures, and its empirical efficacy. However, these methods only work in offline settings, and assumes complete observability of labels in the source domain. Therefore they are not applicable to our online bandit settings.

**Distribution Shift in Bandits.** A related study addresses the challenge of policy learning in contextual bandits using historical observational data, particularly when there is a shift in the environment (Si et al., 2023). The authors propose a distributionally robust approach to policy evaluation and learning, ensuring that the learned policy remains effective even under worst-case environmental shifts. However, their focus is distinct from ours. Their goal is to ensure policy robustness against environmental shifts between training and deployment within the same domain, whereas our proposed DABand focuses on adapting policies from a source domain to a different target domain, particularly when direct feedback from the target domain is limited or costly.

**Domain Adaptation Related to Bandits.** There is also work related to both domain adaptation and bandits. Specifically, Guo et al. (2020) propose a domain adaptation method using a bandit algorithm to select which domain to use during training.

We note that their goal and setting is different from our DABand's. (1) Guo et al. (2020) focuses on improving accuracy in a typical, offline domain adaptation setting. In contrast, our DABand focuses on minimizing regret in an online bandit setting. (2) Guo et al. (2020) assumes complete access to ground-truth labels in the source domain, while in our bandit setting, ground-truth may still be unknown if the action is incorrect. Therefore their work is not applicable to our setting.

## 3 THEORY

In this section, we formalize the problem of contextual bandits across domains and derive the corresponding regret bound. We then develop our DABand inspired by this bound in Sec. 4. **All proofs of lemmas, theorems can be found in the Appendix.**

**Notation.** We use $[k]$ to denote the set $\{1, 2, \cdots, k\}$, for $k \in \mathbb{N}^+$. The Euclidean norm of a vector $\mathbf{x} \in \mathbb{R}^d$ is denoted as $\|\mathbf{x}\|_2 = \sqrt{\mathbf{x}^T \mathbf{x}}$. We denote the operator norm and Frobenius norm of a matrix $\mathbf{W} \in \mathbb{R}^{m \times n}$ as $\|W\|$ and $\|W\|_F$, respectively. Given a semi-definite matrix $A \in \mathbb{R}^{d \times d}$ and a vector $\mathbf{x} \in \mathbb{R}^d$, we denote the Mahalanobis norm as $\|\mathbf{x}\|_A = \sqrt{\mathbf{x}^T A \mathbf{x}}$. For a function $f(T)$ of a parameter $T$, we denote as $\mathcal{O}(f(T))$ the terms growing in the order of $f(T)$, ignoring constant factors. We assume all the action spaces are identical and with cardinality of $K$, i.e., $|\mathcal{A}_1| = |\mathcal{A}_2| = \cdots = |\mathcal{A}_N| = K$. We use $\langle \cdot, \cdot \rangle$ to denote the inner product of two vectors. We denote as $a_{\mathcal{D},i}$ the action $i$ in domain $\mathcal{D}$, and omit the subscript $\mathcal{D}$ when the context is clear, e.g., $x_{i,a_i^*}^{\mathcal{D}} \equiv x_{i,a_{\mathcal{D},i}^*}^{\mathcal{D}}$.

### 3.1 PRELIMINARIES

**Typical Contextual Bandit Setting.** Suppose we have $N$ samples from an unknown domain $\mathcal{D}$, which is represented as $\{\mathbf{x}_{i,a}^{\mathcal{D}}, a_i^*\}_{i \in [N], a \in [K]}$. Here, $\mathbf{x}_{i,a} \in \mathcal{D}$ denotes the context for action $a \in \mathcal{A}_i$, and $a_i^* \in \mathcal{A}_i$ is the ground-truth action that will receive the optimal reward. We denote as $r_{i,a}$ the received reward in round $i$ after performing action $a$ and $\widehat{a}_i \in \mathcal{A}_i$ the chose action by a bandit algorithm in round $i$. The goal in typical contextual bandits is to learn a policy of choosing actions $\widehat{a}_i$ in each round to minimize the regret after $N$ rounds:

$$R = \sum_{i=1}^{N} r_{i,a_i^*} - \sum_{i=1}^{N} r_{i,\widehat{a}_i}. \tag{1}$$

**LinUCB.** LinUCB (Li et al., 2010) is a classic method for the typical contextual bandit setting. It assumes that the reward is linear to its input context, i.e., $\mathbf{x}_{i,a}$. During each round $i$, the agent selects an arm according to historical contexts and their corresponding rewards, $r_{i,a}$:

$$\widehat{a}_i = \underset{a \in \mathcal{A}_i}{\operatorname{argmax}} \left\{ \mathbf{x}_{i,a}^T \theta_i + \alpha \|\mathbf{x}_{i,a}\|_{A_{i-1}^{-1}} \right\},$$

where $A_{i-1} = \gamma \mathbf{I}_d + \sum_{j=1}^{i-1} \mathbf{x}_{j,\widehat{a}_j} \mathbf{x}_{j,\widehat{a}_j}^T$ with $\gamma > 0$ and $\mathbf{x}_{j,\widehat{a}_j}^T$ as the historical selected action's context, $\mathbf{x}_{i,a}$ is the context for the candidate action $a$ in round $i$, $\theta_i$ is the bandit parameter in round $i$ and $\alpha > 0$ is a hyperparameter that adjusts the exploration rate.

**LinUCB with Representation Learning.** In this paper, we use Neural LinUCB (Xu et al., 2020; Wang et al., 2022) as a backbone model to handle high-dimensional context with representation learning. We assume that there exist a ground-truth encoder $\phi^*$ to encode the raw context $\mathbf{x}_{i,a}$ into a latent-space representation (encoding) $\phi^*(\mathbf{x}_{i,a})$. Subsequently, the reward for the context $\mathbf{x}_{i,a}$ is then $r(\mathbf{x}_{i,a}) = \langle \theta^*, \phi^*(\mathbf{x}_{i,a}) \rangle$ plus some stochastic noise $\epsilon$. Furthermore, we use the same setting as in Xu et al. (2020) where the ground-truth rewards are restricted to the range of $[0, 1]$. Additionally, we further bound predicted rewards by normalizing $\|\widehat{\theta}\| = 1$ and $\|\widehat{\phi}(\mathbf{x}_{i,a})\| = 1$.

In the single-domain typical setting, the goal is to learn an encoder $\widehat{\phi}$ and the contextual bandit parameter $\widehat{\theta}$, such that our estimated reward $\widehat{r}(\mathbf{x}_{i,a}) = \langle \widehat{\theta}, \widehat{\phi}(\mathbf{x}_{i,a}) \rangle$ can be close to the ground-truth reward $r(\mathbf{x}_{i,a})$. One can then use this to form derive policies to minimize the regret in Eqn. (1).

For simplicity, in Definition 3.1 below, we further denote as $f_{\mathcal{D}}$ the (ground-truth) labeling function for domain $\mathcal{D}$ and as $h \in \mathcal{H}$ a hypothesis such that $f$ is parameterized by $\phi_{\mathcal{D}}^*, \theta_{\mathcal{D}}^*$, and $h$ is parameterized by $\widehat{\phi}, \widehat{\theta}$.

### 3.2 OUR CROSS-DOMAIN CONTEXTUAL BANDIT SETTING

Typical contextual bandits operate in a single domain $\mathcal{D}$. In contrast, our cross-domain bandit setting involves a source domain $\mathcal{S}$ and a target domain $\mathcal{T}$. In this setting, one can only collect feedback (reward) from the source domain, but not from the target domain. Specifically: (1) We assume a low-cost *source domain* (experiments on mice), where for each round $i$, one has access

to the contexts $\{\mathbf{x}_{i,a}^{\mathcal{S}}\}_{a \in [K]}$ for each candidate actions, chooses one action $\widehat{a}_i$, and receives reward $r_{i,\widehat{a}_i}^{\mathcal{S}}$. (2) Additionally, we assume a high-cost *target domain* where collecting feedback (reward) is expensive (experiments on humans); therefore, one only has access to the contexts $\{\mathbf{x}_{i,a}^{\mathcal{T}}\}_{a \in [K]}$ for each candidate actions for each round, but **cannot collect feedback (reward) $r_{i,\widehat{a}_i}^{\mathcal{T}}$ for any action in the target domain**. The goal in our cross-domain contextual bandits is to learn a policy of choosing actions $\widehat{a}_i$ in the target domain to minimize the target-domain **zero-shot regret** for $N$ (hypothetical) future rounds:

$$R_{\mathcal{T}} = \sum_{i=1}^{N} r_{i,a_i^*}^{\mathcal{T}} - \sum_{i=1}^{N} r_{i,\widehat{a}_i}^{\mathcal{T}}. \tag{2}$$

**Difference between Eqn. (1) and Eqn. (2).** In the typical setting, Eqn. (1) uses different updated policies for each round; this is also true for the source domain. In contrast, the target regret in Eqn. (2) use the same fixed policy obtained from the source domain for all $N$ (hypothetical) rounds because one cannot collect feedback from the target domain. See the difference between Definition 3.4 and Definition 3.5 below for a more formal comparison.

### 3.3 FORMAL DEFINITIONS OF ERROR

**Definition 3.1 (Labeling Function and Hypothesis).** *We define the labeling function $f_{\mathcal{D}}$ and the hypothesis $h$ for domain $\mathcal{D}$ as follows:*

$$f_{\mathcal{D}}(\mathbf{x}_{i,a}^{\mathcal{D}}) = r(\mathbf{x}_{i,a}^{\mathcal{D}}) = \langle \theta_{\mathcal{D}}^*, \phi_{\mathcal{D}}^*(\mathbf{x}_{i,a}^{\mathcal{D}}) \rangle + \epsilon_i$$

$$h(\mathbf{x}_{i,a}^{\mathcal{D}}) = \widehat{r}(\mathbf{x}_{i,a}^{\mathcal{D}}) = \langle \widehat{\theta}, \widehat{\phi}(\mathbf{x}_{i,a}^{\mathcal{D}}) \rangle + \epsilon_i$$

*where $\theta_{\mathcal{D}}^*$ is the optimal predictor for domain $\mathcal{D}$, $\widehat{\theta}$ denotes the estimated predictor, and $\epsilon_i$ is the random noise.*

Next, we analyze how well our hypothesis $h$ estimates the labeling function $f$ (i.e., ground-truth hypothesis). Definition 3.2 below quantifies the closeness between two hypotheses by calculating the absolute difference in their estimated rewards.

**Definition 3.2 (Estimated Error between Two Hypotheses).** *Assuming all contexts $\mathbf{x}_{i,\widehat{a}_i}^{\mathcal{D}}$ are from domain $\mathcal{D}$, the error between two hypotheses $h_1, h_2 \in \mathcal{H}$ on domain $\mathcal{D}$ given selected actions $\{\widehat{a}_i\}_{i \in [N]} \in [K]$ is*

$$\epsilon_{\mathcal{D}}(h_1, h_2) = \sum_{i=1}^{N} \left( \left| h_1(\mathbf{x}_{i,\widehat{a}_i}^{\mathcal{D}}) - h_2(\mathbf{x}_{i,\widehat{a}_i}^{\mathcal{D}}) \right| \right) = \sum_{i=1}^{N} \left( \left| \langle \widehat{\theta}_1, \widehat{\phi}_1(\mathbf{x}_{i,\widehat{a}_i}^{\mathcal{D}}) \rangle - \langle \widehat{\theta}_2, \widehat{\phi}_2(\mathbf{x}_{i,\widehat{a}_i}^{\mathcal{D}}) \rangle \right| \right). \tag{3}$$

Note that the domain $\mathcal{D}$ above can be the source domain $\mathcal{S}$ or the target domain $\mathcal{T}$. We then define the error of our estimated hypothesis in these domains.

**Definition 3.3 (Source- and Target-Domain Error).** *With $N$ samples from the source domain $\mathcal{S}$, and $f_S$ denoting the labeling function for $\mathcal{S}$, the source-domain error is then*

$$\epsilon_{\mathcal{S}}(f_{\mathcal{S}}, h) = \sum_{i=1}^{N} \left( \left| f_S(\mathbf{x}_{i,\widehat{a}_i}^{\mathcal{S}}) - h(\mathbf{x}_{i,\widehat{a}_i}^{\mathcal{S}}) \right| \right) = \sum_{i=1}^{N} \left( \left| \langle \theta_{\mathcal{S}}^*, \phi_{\mathcal{S}}^*(\mathbf{x}_{i,\widehat{a}_i}^{\mathcal{S}}) \rangle - \langle \widehat{\theta}, \widehat{\phi}(\mathbf{x}_{i,\widehat{a}_i}^{\mathcal{S}}) \rangle \right| \right),$$

*where $\mathbf{x}_{i,\cdot}^{\mathcal{S}}$ comes from $\mathcal{S}$. Furthermore, $\widehat{a}_i = \arg\max_a h(\mathbf{x}_{i,a}^{\mathcal{S}})$ and $a_i^* = \arg\max_a f_S(\mathbf{x}_{i,a}^{\mathcal{S}})$. For simplicity, we shorten the notation $\epsilon_{\mathcal{S}}(f_{\mathcal{S}}, h)$ to $\epsilon_{\mathcal{S}}(h)$ and similarly use $\epsilon_{\mathcal{T}}(h)$ for domain $\mathcal{T}$. Assuming $\mathbf{x}_{i,\cdot}^{\mathcal{T}}$ comes from $\mathcal{T}$, the estimated error for the target domain is then:*

$$\epsilon_{\mathcal{T}}(f_{\mathcal{T}}, h) = \sum_{i=1}^{N} \left( \left| f_{\mathcal{T}}(\mathbf{x}_{i,\widehat{a}_i}^{\mathcal{T}}) - h(\mathbf{x}_{i,\widehat{a}_i}^{\mathcal{T}}) \right| \right) = \sum_{i=1}^{N} \left( \left| \langle \theta_{\mathcal{T}}^*, \phi_{\mathcal{T}}^*(\mathbf{x}_{i,\widehat{a}_i}^{\mathcal{T}}) \rangle - \langle \widehat{\theta}, \widehat{\phi}(\mathbf{x}_{i,\widehat{a}_i}^{\mathcal{T}}) \rangle \right| \right).$$

### 3.4 SOURCE REGRET AND TARGET REGRET

Below define the regret for the source and target domains.

**Definition 3.4 (Source Regret).** *Assuming $\mathbf{x}_{i,\cdot}^{\mathcal{S}}$ comes from domain $\mathcal{S}$, the source regret (i.e., the regret in the source domain) is*

$$R_{\mathcal{S}} = \sum_{i=1}^{N} \left( f_S(\mathbf{x}_{i,a_i^*}^{\mathcal{S}}) - f_S(\mathbf{x}_{i,\widehat{a}_i}^{\mathcal{S}}) \right) = \sum_{i=1}^{N} \left( \langle \theta_S^*, \phi_S^*(\mathbf{x}_{i,a_i^*}^{\mathcal{S}}) \rangle - \langle \theta_S^*, \phi_S^*(\mathbf{x}_{i,\widehat{a}_i}^{\mathcal{S}}) \rangle \right). \tag{4}$$

The goal in our cross-domain contextual bandits is to learn a policy of choosing actions $\widehat{a}_i$ in the high-cost target domain (e.g., human experiments) by collecting feedback only in the source domain (e.g., mouse experiments). Formally, we would like to minimize the target regret as defined below.

**Definition 3.5** (**Target Regret and Problem Formulation**). *Denoting the estimated hypothesis as $\widehat{h} = \left\{ \widehat{\phi}, \widehat{\theta} \right\}$, the target regret we aim to minimize is defined as*

$$R_{\mathcal{T}} = \sum_{i=1}^{N} \left( f_{\mathcal{T}}(\mathbf{x}_{i,a_i^*}^{\mathcal{T}}) - f_{\mathcal{T}}(\mathbf{x}_{i,\widehat{a}_i}^{\mathcal{T}}) \right)$$

$$s.t. \ \widehat{a}_i = \underset{a}{\operatorname{argmax}} \ \widehat{h}(\mathbf{x}_{i,a}^{\mathcal{T}}) + \alpha \|\widehat{\phi}(\mathbf{x}_{i,a}^{\mathcal{T}})\|_{[A^{\mathcal{S}}]^{-1}}, \quad \widehat{h} = \underset{h}{\operatorname{argmin}} \ \epsilon_{\mathcal{S}}(h),$$

*where $A^{\mathcal{S}} = \gamma \mathbf{I} + \sum_{i=1}^{N} [\widehat{\phi}(\mathbf{x}_{i,\widehat{a}_i}^{\mathcal{S}})][\widehat{\phi}(\mathbf{x}_{i,\widehat{a}_i}^{\mathcal{S}})]^T$ denotes the context matrix accumulated by the selected context features in the source domain.*

## 3.5 CROSS-DOMAIN ERROR BOUND FOR REGRESSION

Prior to introducing our final regret bound, it is necessary to define an additional component. A fundamental challenge in general domain adaptation problems is to manage the divergence between source and target domains. Unfortunately, previous domain adaptation theory only covers *classification* tasks (Ben-David et al., 2010; Zhang et al., 2019). In contrast, the problem of contextual bandits is essentially a reward *regression* problem. To address this challenge, we introduce the following new definition to bound the error between the source and target domains.

**Definition 3.6** ($\mathcal{H}\Delta\mathcal{H}$ **Hypothesis Space for Regression**). *For a hypothesis space $\mathcal{H}$, the symmetric difference hypothesis space $\mathcal{H}\Delta\mathcal{H}$ is the set of hypotheses s.t.*

$$g \in \mathcal{H}\Delta\mathcal{H} \quad \Longleftrightarrow \quad g(\mathbf{x}) = |h(\mathbf{x}) - h'(\mathbf{x})|.$$

*We then can further define the Divergence for $\mathcal{H}\Delta\mathcal{H}$ Hypothesis Space:*

$$\widehat{d}_{\mathcal{H}\Delta\mathcal{H}}(\mathcal{S}, \mathcal{T}) = 2 \sup_{h,h' \in \mathcal{H}} |\mathbb{E}_{\mathbf{x} \sim \mathcal{S}}[|h(\mathbf{x}) - h'(\mathbf{x})|] - \mathbb{E}_{x \sim \mathcal{T}}[|h(\mathbf{x}) - h'(\mathbf{x})|]|$$

**Definition 3.7** (**Optimal Hypothesis**). *The optimal hypothesis, denoted as $h^*$, can balance the estimated error for both source and target domain. Formally,*

$$h^* = \arg\min_{h \in \mathcal{H}} \ \epsilon_{\mathcal{S}}(h) + \epsilon_{\mathcal{T}}(h),$$

*and we define the minimum estimated error for the optimal hypothesis $h^*$ as*

$$\psi = \epsilon_{\mathcal{S}}(h^*) + \epsilon_{\mathcal{T}}(h^*).$$

## 3.6 FINAL REGRET BOUND

With all the definitions of source/target regret in Definition 3.4 and Definition 3.5, we can then bound the target regret using Theorem 3.1 below.

**Theorem 3.1** (**Target Regret Bound**). *Denoting the contexts from the source domain $\mathcal{S}$ as $\{\mathbf{x}_{t,a}^{\mathcal{S}}\}_{i \in [N], a \in [K]}$, the associated ground-truth action as $\{a_i^*\}_{i=1}^{N}$, and the contexts from the target domain $\mathcal{T}$ as $\{\mathbf{x}_{t,a}^{\mathcal{T}}\}_{i \in [N], a \in [K]}$, the upper bound for our target regret $R_{\mathcal{T}}$ is*

$$R_{\mathcal{T}} \triangleq \sum_{i=1}^{N} \left( \left| \langle \theta_{\mathcal{T}}^*, \phi_{\mathcal{T}}^*(\mathbf{x}_{i,a_i^*}^{\mathcal{T}}) \rangle - \langle \theta_{\mathcal{T}}^*, \phi_{\mathcal{T}}^*(\mathbf{x}_{i,\widehat{a}_i}^{\mathcal{T}}) \rangle \right| \right)$$

$$\leq \underbrace{R_{\mathcal{S}}}_{Source\ Regret} + \underbrace{2 \cdot \epsilon_{\mathcal{S}}(h)}_{Regression\ Error} + \underbrace{N \cdot \widehat{d}_{\mathcal{H}\Delta\mathcal{H}}(\mathcal{S}, \mathcal{T})}_{Data\ Divergence} + \underbrace{\psi + C}_{Constant}$$

$$+ \underbrace{\sum_{i=1}^{N} \left( \left| \langle \widehat{\theta}, \widehat{\phi}(\mathbf{x}_{i,\widehat{a}_i}^{\mathcal{T}}) \rangle \right| \right) + \sum_{i=1}^{N} \mathbb{1}[a_i^* \neq \widehat{a}_i] \left( \left| \langle \widehat{\theta}, \widehat{\phi}(\mathbf{x}_{i,\widehat{a}_i}^{\mathcal{S}}) \rangle \right| \right)}_{Predicted\ Rewards}, \quad (5)$$

*where $R_{\mathcal{S}}$, $\epsilon_{\mathcal{S}}(h)$, and $\widehat{d}_{\mathcal{H}\Delta\mathcal{H}}$ are the source regret, source-domain error, and $\mathcal{H}\Delta\mathcal{H}$ divergence defined in Definition 3.4, Definition 3.3, and Definition 3.6, respectively. $\psi$ is a constant independent to the problem and and $C$ is a constant which can be ignored (see the Appendix for more details) .*

Here we provide several observations (for more analysis, please **refer to the Appendix**):

(1) Since $R_{\mathcal{S}}$ is sub-linear w.r.t. the number of samples in the source domain, so is $R_{\mathcal{T}}$.
(2) Theorem 3.1 bounds the target regret using the source regret, enabling the exploration of the target domain by only collecting feedback (reward) from the source domain.
(3) Typical generalization bounds in DA naively minimize the source regret $R_{\mathcal{S}}$ while aligning source and target data in the latent space (i.e., minimizing $\widehat{d}_{\mathcal{H}\Delta\mathcal{H}}$). This is **not sufficient**, as we need two additional terms, i.e., the regression error $\epsilon_{\mathcal{S}}(h)$ and the predicted reward $\sum_{i=1}^{N}\left(\left|\langle\widehat{\theta},\widehat{\phi}(\mathbf{x}_{i,\widehat{a}_i}^{\mathcal{T}})\rangle\right|\right) + \sum_{i=1}^{N}\mathbb{1}[a_i^* \neq \widehat{a}_i]\left(\left|\langle\widehat{\theta},\widehat{\phi}(\mathbf{x}_{i,\widehat{a}_i}^{\mathcal{S}})\rangle\right|\right)$, as regularization terms.
(4) Minimizing the regression error $\epsilon_{\mathcal{S}}(h)$ encourages accurate prediction of source rewards.
(5) Minimizing the predicted reward $\sum_{i=1}^{N}\left(\left|\langle\widehat{\theta},\widehat{\phi}(\mathbf{x}_{i,\widehat{a}_i}^{\mathcal{T}})\rangle\right|\right) + \sum_{i=1}^{N}\mathbb{1}[a_i^* \neq \widehat{a}_i]\left(\left|\langle\widehat{\theta},\widehat{\phi}(\mathbf{x}_{i,\widehat{a}_i}^{\mathcal{S}})\rangle\right|\right)$ regularizes the model to avoid overestimating rewards.

Note that Theorem 3.1 is **nontrivial**. While it does resemble the generalization bound in domain adaptation, there are key differences. As mentioned in Observation (3) above, our target regret bound includes two additional crucial terms not found in domain adaptation. Specifically:

- **Regression Error in the Source Domain.** $\sum_{i=1}^{N}\left(\left|\langle\theta_{\mathcal{S}}^*,\phi_{\mathcal{S}}^*(x_{i,\widehat{a}_i}^{\mathcal{S}})\rangle - \langle\widehat{\theta},\widehat{\phi}(x_{i,\widehat{a}_i}^{\mathcal{S}})\rangle\right|\right)$ (i.e.,
  $\epsilon_{\mathcal{S}}(h)$ in Theorem 3.1) defines the difference between the true reward from selecting action $\widehat{a}_i$ and the estimated reward for this action.
- **Predicted Reward.** $\sum_{i=1}^{N}\left(\left|\langle\widehat{\theta},\widehat{\phi}(\mathbf{x}_{i,\widehat{a}_i}^{\mathcal{T}})\rangle\right|\right) + \sum_{i=1}^{N}\mathbb{1}[a_i^* \neq \widehat{a}_i]\left(\left|\langle\widehat{\theta},\widehat{\phi}(\mathbf{x}_{i,\widehat{a}_i}^{\mathcal{S}})\rangle\right|\right)$ serves as a regularization term to regularize the model to avoid overestimating rewards.

The results of the ablation study in Table 4 in Sec. 5 highlight the significance of these two terms. See Appendix G.4 for more discussion on novelty as well as key differences between our DABand and classic domain adaptation.

## 4 METHOD

With Theorem 3.1, we can then design our DABand algorithm to obtain optimal target regret. We discuss how to translate each term in Eqn. (5) to a differential loss term in Sec. 4.1 and then put them together in Sec. 4.2.

### 4.1 FROM THEORY TO PRACTICE: TRANSLATING THE BOUND IN EQN. (5) TO DIFFERENTIABLE LOSS TERMS

**Source Regret.** Inspired by Neural-LinUCB (Xu et al., 2020), we use LinUCB (Li et al., 2010) to update $\widehat{\theta}$ and the following empirical loss function when updating the encoder parameters $\widehat{\phi}$ by back-propagation in round $i$:

$$\mathcal{L}_i^{R_{\mathcal{S}}} = \sum_{a=1}^{K}\left(\widehat{\theta}^T\widehat{\phi}(\mathbf{x}_{i,a}) - r(\mathbf{x}_{i,a})\right)^2, \tag{6}$$

where $\theta$ is the contextual bandit parameter, and $\widehat{\phi}$ is the encoder shared by the source and target domains, i.e., we set $\widehat{\phi}_{\mathcal{S}} = \widehat{\phi}_{\mathcal{T}}$ during training.

**Insights.** From Eqn. (6), a fundamental difference between classification problems and bandit problems becomes apparent. In classification tasks, both the ground-truth label and the estimated label are accessible. However, in bandit problems, we only know the estimated label and reward. If the estimated rewards align with our estimated label's correctness, it suggests knowledge of the ground-truth label for that feature. If not, we can only deduce that the estimated label is inaccurate, without identifying the correct label among the other candidates. This uncertainty significantly amplifies the complexity of contextual bandit problems.

**Regression Error.** We use $L_1$ loss to optimize the regression error, defined as the difference between the true reward from selecting action $\widehat{a}_i$ and the estimated reward for this action (see Definition 3.3 for a detailed explanation of this regression error). Minimizing the *regression error* term directly

---

**Algorithm 1** DABand Training Algorithm

---

1: **Input:** regularization parameter $\gamma > 0$, number of total steps $N$, episode length $H$, exploration parameters $\alpha$.
2: **Output:** parameters of the model: $\widehat{\phi}_N, \widehat{\theta}_N, A_N, b_N$.
3: **Initialization:** $A_0 = \gamma \mathbf{I}, b_0 = \mathbf{0}$, and for $\widehat{\theta}_0$ and $\widehat{\phi}_0$ are initialized following (Xu et al., 2020).
4: **for** $i = 1, ..., N$ **do**
5:  Obtain $\left\{\mathbf{x}_{i,a}^{\mathcal{S}}\right\}_{a \in [K]}$ and $\left\{\mathbf{x}_{i,a}^{\mathcal{T}}\right\}_{a \in [K]}$ from the source and the target domain, respectively.
6:  Choose an action $\widehat{a}_i = \operatorname{argmax}_{a \in [K]} \left[\widehat{\phi}_{i-1}(\mathbf{x}_{i,a}^{\mathcal{S}})\right] \widehat{\theta}_{i-1} + \alpha \left\|\left[\widehat{\phi}_{i-1}(\mathbf{x}_{i,a}^{\mathcal{S}})\right]\right\|_{A_{i-1}^{-1}}$.
7:  Get the reward $r_i = \mathbf{r}(\mathbf{x}_{i,\widehat{a}_i})$ based on the selected action $\widehat{a}_i$.
8:  Update bandit parameters:
9:  $\begin{cases} A_i = A_{i-1} + \left[\widehat{\phi}_{i-1}(\mathbf{x}_{i,\widehat{a}_i}^{\mathcal{S}})\right]\left[\widehat{\phi}_{i-1}(\mathbf{x}_{i,\widehat{a}_i}^{\mathcal{S}})\right]^T, \\ b_i = b_{i-1} + r_i \left[\widehat{\phi}_{i-1}(\mathbf{x}_{i,\widehat{a}_i}^{\mathcal{S}})\right], \\ \widehat{\theta}_i = A_i^{-1} b_i \end{cases}$
10:  **if** $\mod (i, H) = 0$ **then**
11:    **for** $j = 1, ..., H$ **do**
12:      Calculate $\mathcal{L}^{DABand}$ in Eqn. (11).
13:      Use the Adam optimizer (Diederik, 2014) to update encoder $\widehat{\phi}_i$ and discriminator $g$ by back-propagation to solve the minimax optimization in Eqn. (10).
14:    **end for**
15:  **else**
16:    $\widehat{\phi}_i = \widehat{\phi}_{i-1}$
17:  **end if**
18: **end for**
19: **Output:** $\widehat{\phi}_N, \widehat{\theta}_N, A_N, b_N$.

---

*encourages accurate prediction of source rewards* using $L_1$ loss. Specifically, we use

$$\mathcal{L}_i^{\epsilon_{\mathcal{S}}(h)} = \left| \langle \theta_{\mathcal{S}}^*, \phi_{\mathcal{S}}^*(\mathbf{x}_{i,\widehat{a}_i}^{\mathcal{S}}) \rangle - \langle \widehat{\theta}, \widehat{\phi}(\mathbf{x}_{i,\widehat{a}_i}^{\mathcal{S}}) \rangle \right|, \tag{7}$$

where $\langle \theta_{\mathcal{S}}^*, \phi_{\mathcal{S}}^*(\mathbf{x}_{i,\widehat{a}_i}^{\mathcal{S}}) \rangle$ is the reward received by the agent in the source domain. Minimizing the regression error term above directly encourages accurate prediction of source rewards.

**Data Divergence.** The data divergence term in Eqn. (5) leads to following loss term:

$$\mathcal{L}^{div} = \max_g N \cdot (\mathbb{E}^{\mathcal{S}}[\mathcal{L}_D(g(\widehat{\phi}(\mathbf{x})), 0)] + \mathbb{E}^{\mathcal{T}}[\mathcal{L}_D(g(\widehat{\phi}(\mathbf{x})), 1)]), \tag{8}$$

where $\mathbb{E}^{\mathcal{S}}$ and $\mathbb{E}^{\mathcal{T}}$ denote expectations over the data distributions of $(\mathbf{x}, a)$ in the source and target domains, respectively. $g$ is a discriminator that classifies whether $\mathbf{x}$ is from the source domain or the target domain. $\mathcal{L}_D$ is the binary classification accuracy, where labels 0 and 1 indicate the source and target domains, respectively. In practice, we use the cross-entropy loss as a differentiable surrogate loss. As in Ganin et al. (2016b), solving the minimax optimization $\min_{\widehat{\phi}} \max_g \mathcal{L}^{div}$ is equivalent to aligning source- and target-domain data distributions in the latent (encoding) space induced by the encoder $\widehat{\phi}$.

**Predicted Reward.** The predicted reward term in Eqn. (5), i.e., $\sum_{i=1}^{N} \left( \left| \langle \widehat{\theta}, \widehat{\phi}(\mathbf{x}_{i,\widehat{a}_i}^{\mathcal{T}}) \rangle \right| \right) + \sum_{i=1}^{N} \mathbb{1}[a_i^* \neq \widehat{a}_i] \left( \left| \langle \widehat{\theta}, \widehat{\phi}(\mathbf{x}_{i,\widehat{a}_i}^{\mathcal{S}}) \rangle \right| \right)$, can be translated to the loss term $\sum_{i=1}^{N} \mathcal{L}_i^{reg}$, where for each round $i$ we have

$$\mathcal{L}_i^{reg} = \left| \langle \widehat{\theta}, \widehat{\phi}(\mathbf{x}_{i,\widehat{a}_i}^{\mathcal{T}}) \rangle \right| + \mathbb{1}[a_i^* \neq \widehat{a}_i] \left( \left| \langle \widehat{\theta}, \widehat{\phi}(\mathbf{x}_{i,\widehat{a}_i}^{\mathcal{S}}) \rangle \right| \right). \tag{9}$$

Minimizing the predicted reward $\mathcal{L}_i^{reg}$ regularizes the model to avoid overestimating rewards.

**Constant Term.** In Eqn. (5), $\psi + C$ is the constant term, where $\psi$ is defined in Definition 3.7, and $C$ contains some small constants which can be ignored (see the Appendix for more details).

Table 1: Accuracy on the target domain for the DIGIT dataset. Note that in this zero-shot target regret setting, the accuracy $ACC = 1 - \frac{1}{N}R_{\mathcal{T}}$, where $R_{\mathcal{T}}$ is the target regret.

| Metrics | Acc Per Class ↑ | | | | | | | | | | Acc ↑ |
|---|---|---|---|---|---|---|---|---|---|---|---|
| Test Accuracy | 0 | 1 | 2 | 3 | 4 | 5 | 6 | 7 | 8 | 9 | Average |
| LinUCB (Li et al., 2010) | 0.3667±0.03 | 0.4045±0.02 | 0.5102±0.03 | 0.3811±0.01 | 0.3157±0.04 | 0.3445±0.01 | 0.4229±0.03 | 0.4595±0.03 | 0.3489±0.02 | 0.2449±0.01 | 0.3808±0.02 |
| LinUCB-P (Li et al., 2010) | 0.3645±0.02 | 0.6398±0.04 | 0.3151±0.02 | 0.2709±0.03 | 0.2258±0.01 | 0.1995±0.02 | 0.1752±0.01 | 0.3786±0.04 | 0.1273±0.02 | 0.2992±0.03 | 0.3060±0.02 |
| NLinUCB (Xu et al., 2020) | 0.3781±0.04 | 0.3228±0.03 | 0.3569±0.02 | 0.3381±0.04 | 0.3663±0.04 | 0.4312±0.05 | 0.4766±0.05 | 0.4004±0.05 | 0.4034±0.02 | 0.3613±0.03 | 0.3816±0.04 |
| NLinUCB-P (Xu et al., 2020) | **0.8588**±0.03 | 0.2224±0.02 | 0.3880±0.03 | 0.3029±0.02 | 0.2978±0.03 | 0.0000±0.01 | 0.1869±0.02 | 0.2243±0.01 | 0.2148±0.01 | 0.0000±0.00 | 0.2706±0.02 |
| DABand (OURS) | 0.6891±0.10 | **0.6662**±0.03 | **0.6188**±0.01 | **0.5703**±0.05 | **0.6008**±0.05 | **0.5534**±0.03 | **0.6010**±0.02 | **0.5709**±0.01 | **0.6266**±0.04 | **0.5023**±0.02 | **0.6002**±0.02 |
| **OURS** VS. NLINUCB | **+0.3110** | **+0.3434** | **+0.2619** | **+0.2322** | **+0.2345** | **+0.1222** | **+0.1244** | **+0.1705** | **+0.2232** | **+0.1410** | **+0.2186** |

Table 2: Accuracy on the target domain for the VisDA17 dataset. Note that in this zero-shot target regret setting, the accuracy $ACC = 1 - \frac{1}{N}R_{\mathcal{T}}$, where $R_{\mathcal{T}}$ is the target regret.

| Metrics | Acc Per Class ↑ | | | | | | | | | | | | Acc ↑ |
|---|---|---|---|---|---|---|---|---|---|---|---|---|---|
| Test Accuracy | aeroplane | bicycle | bus | car | horse | knife | motorcycle | person | plant | skateboard | train | truck | Average |
| LinUCB (Li et al., 2010) | N/A | N/A | N/A | N/A | N/A | N/A | N/A | N/A | N/A | N/A | N/A | N/A | N/A |
| LinUCB-P (Li et al., 2010) | 0.0510±0.02 | 0.0190±0.00 | 0.1360±0.05 | 0.0440±0.02 | 0.0390±0.01 | **0.5580**±0.06 | 0.0410±0.02 | 0.2050±0.05 | 0.0850±0.04 | 0.0920±0.13 | 0.0580±0.02 | 0.0460±0.12 | 0.1145±0.06 |
| NLinUCB (Xu et al., 2020) | 0.2870±0.03 | 0.0220±0.02 | 0.0000±0.00 | 0.1250±0.02 | 0.0000±0.00 | 0.0130±0.02 | 0.0470±0.02 | 0.0000±0.00 | 0.0460±0.02 | 0.0050±0.00 | 0.6550±0.04 | 0.0000±0.00 | 0.1001±0.02 |
| NLinUCB-P (Xu et al., 2020) | 0.0060±0.01 | 0.0020±0.01 | **0.9222**±0.02 | 0.0010±0.00 | 0.0080±0.01 | 0.0990±0.03 | 0.0030±0.00 | 0.0020±0.00 | 0.0180±0.02 | 0.0420±0.01 | 0.0000±0.00 | 0.0150±0.01 | 0.0932±0.01 |
| DABand (OURS) | **0.5504**±0.05 | **0.3562**±0.04 | 0.4462±0.02 | **0.3844**±0.01 | **0.5582**±0.04 | 0.2632±0.03 | **0.6474**±0.03 | **0.2396**±0.03 | **0.4882**±0.04 | **0.4062**±0.02 | **0.7685**±0.05 | **0.1968**±0.02 | **0.4644**±0.03 |
| **OURS** VS. NLINUCB | **+0.2634** | **+0.3342** | **+0.4462** | **+0.2594** | **+0.5582** | **+0.2502** | **+0.6004** | **+0.2396** | **+0.4422** | **+0.4012** | **+0.1135** | **+0.1968** | **+0.3643** |

## 4.2 PUTTING EVERYTHING TOGETHER: DABAND TRAINING ALGORITHM

Putting all non-constant loss terms above together, we have the final minimax optimization problem

$$\min_{\widehat{\phi}} \max_g \mathcal{L}^{DABand}, \tag{10}$$

where the value function (objective function):

$$\mathcal{L}^{DABand} = \sum_{i=1}^{N}(\mathcal{L}_i^{R_{\mathcal{S}}} + 2 \cdot \mathcal{L}_i^{\epsilon_{\mathcal{S}}(h)} + \mathcal{L}_i^{reg}) + \lambda \cdot \mathcal{L}_i^{div}. \tag{11}$$

Note that we need to alternate between updating $\widehat{\theta}$ using LinUCB Li et al. (2010) and updating the encoder $\widehat{\phi}$ with the discriminator $g$ in Eqn. (10).

Formally, our method, described in Alg. 1, operates as follows: In each iteration, we access the context in the original feature space (i.e., $\mathbb{R}^d$) for each action from both source and target domains, denoted as $\left\{x_{i,a}^{\mathcal{S}}\right\}_{a\in[K]}$ and $\left\{x_{i,a}^{\mathcal{T}}\right\}_{a\in[K]}$ respectively. Subsequently, we compute the latent representation using the encoder $\phi(\cdot)$ and select the action $\widehat{a}_i$ for iteration $i$ based on the LinUCB selection rule. The bandit parameters (i.e., $\theta, A, b$) are updated in each iteration. The weights in $\phi$ are updated every episode of length $H$ (i.e., a batch with $H$ past iterations) using our objective function in Eqn. (10), following the same rule as in Neural-LinUCB (Xu et al., 2020).

## 5 EXPERIMENTS

In this section, we compare DABand with existing methods on real-world datasets.

**Datasets.** To demonstrate the effectiveness of our DABand, We evaluate our methods in terms of prediction accuracy and **zero-shot** target regret on three datasets, i.e., DIGIT (Ganin et al., 2016a), VisDA17 (Peng et al., 2017), and S2RDA49 (Tang & Jia, 2023). See details for each dataset in Appendix B.

**Baselines.** We compare our DABand with both classic and state-of-the-art contextual bandit algorithms, including **LinUCB** (Li et al., 2010), LinUCB with principle component analysis (PCA) pre-processing (i.e., **LinUCB-P**), Neural-LinUCB (Xu et al., 2020) (**NLinUCB**), and Neural-LinUCB variant that incorporates PCA (i.e., **NLinUCB-P**). Details for each baselines and discussion are in Appendix C. Note that domain adaptation baselines are **not applicable** to our setting because it only works in offline settings and assumes complete observability of labels in the source domain (see Appendix C for details).

**Zero-Shot Target Regret (Accuracy).** We evaluate different methods on the zero-shot target regret setting in Definition 3.5, where all methods can only collect feedback from the source domain, but not the target domain. In this setting, accuracy is equal to 1 minus the average target regret, i.e.,

Table 3: Accuracy on the target domain for the S2RDA49 dataset. Note that in this zero-shot target regret setting, the accuracy $ACC = 1 - \frac{1}{N}R_{\mathcal{T}}$, where $R_{\mathcal{T}}$ is the target regret.

| Metrics | Acc Per Class ↑ | | | | | | | | | | Acc ↑ |
|---|---|---|---|---|---|---|---|---|---|---|---|
| Test Accuracy | aeroplane | bicycle | bus | car | knife | motorcycle | plant | skateboard | train | truck | Average |
| LINUCB (Li et al., 2010) | N/A | N/A | N/A | N/A | N/A | N/A | N/A | N/A | N/A | N/A | N/A |
| LINUCB-P (Li et al., 2010) | 0.1430±0.02 | 0.2600±0.03 | 0.0810±0.01 | 0.0240±0.01 | 0.1070±0.02 | 0.0570±0.01 | 0.1060±0.01 | 0.0390±0.01 | 0.0960±0.02 | 0.1260±0.03 | 0.1039±0.02 |
| NLINUCB (Xu et al., 2020) | 0.0350±0.00 | 0.1210±0.02 | 0.0510±0.01 | 0.0870±0.02 | 0.0270±0.00 | 0.0240±0.01 | 0.2720±0.04 | 0.0090±0.00 | **0.1730**±0.02 | 0.0309±0.01 | 0.1108±0.02 |
| NLINUCB-P (Xu et al., 2020) | 0.0000±0.00 | 0.0000±0.00 | 0.0000±0.00 | **0.2990**±0.03 | **0.7000**±0.02 | 0.0000±0.00 | 0.0000±0.00 | 0.0000±0.00 | 0.0000±0.00 | 0.1160±0.01 | 0.1115±0.01 |
| DABAND (OURS) | **0.5501**±0.03 | **0.6418**±0.03 | **0.5844**±0.03 | 0.2346±0.01 | 0.0410±0.01 | **0.7580**±0.04 | **0.5689**±0.04 | **0.1345**±0.02 | 0.1118±0.02 | **0.3126**±0.02 | **0.3923**±0.03 |
| **OURS** VS. NLINUCB | +0.5151 | +0.5208 | +0.5334 | +0.1476 | +0.0140 | +0.7340 | +0.2969 | +0.1255 | -0.0612 | +0.2817 | +0.2815 |

Figure 1: The cumulative regrets of different methods for 1,920 rounds on DIGIT (**left**), VisDA17 (**middle**), and S2RDA49 (**right**). Results are averaged over 5 runs. LinUCB is not reported for VisDA17 and S2RDA49 due to out-of-memory issues.

$ACC = 1 - \frac{1}{N}R_{\mathcal{T}}$, where $R_{\mathcal{T}}$ is defined in Eqn. (2) and Definition 3.5. Therefore higher accuracy indicates lower target regret and better performance.

We report the per-class accuracy and average accuracy of different methods in Table 1, Table 2 and Table 3 for DIGIT, VisDA17 and S2RDA49, respectively. Our DABand demonstrates favorable performance against NLinUCB, a leading contextual bandit algorithm with representation learning. Despite NLinUCB's superior representation learning capabilities through neural networks (NN) over LinUCB – yielding impressive performance in single-domain applications – its efficacy in cross-domain tasks is heavily contingent on the disparity between the source and target domains. Typically, the strengths of NN, which include enhanced representation learning (feature extraction), may inadvertently amplify this domain divergence, potentially leading to overfitting on the source domain. Conversely, our DABand algorithm not only improves accuracy but also adeptly addresses the challenges posed by domain divergence. Note that LinUCB encounters an out-of-memory issue due to the high-dimensionality of the data in VisDA17 and S2RDA49, marked as "N/A" in Table 2 and Table 3, respectively.

**Performance Analysis of LinUCB and NLinUCB.** Our target regret bound in Eqn. (5) consists of 5 terms, i.e., source regret, regression error, data divergence, constant, and predicted reward. After applying linear contextual bandits on the source domain, the source regret enjoys a sub-linear rate. However, in the linear model, the encoder $\widehat{\phi}(\mathbf{x}) = \mathbf{x}$ is an identity function and therefore fail to align source-domain and target-domain contexts, leading to an unbounded, large data divergence term in Eqn. (5). This is the main reason why linear contextual bandits, such as LinUCB, perform poorly in the cross-domain contextual bandit setting.

Note that even contextual bandits using a nonlinear encoder, e.g., Neural-LinUCB (NLinUCB), fail in this case because their encoders $\widehat{\phi}(\mathbf{x})$ are not learned to align source-domain and target-domain contexts; they therefore will still lead to an unbounded, large data divergence term in Eqn. (5), and subsequently poor performance in the cross-domain contextual bandit setting.

**Continued Training in Target Domains and Cumulative Regret.** Besides zero-shot target regret, we also evaluate how different methods perform when one continues their training in the high-cost target domain and starts to collect feedback (reward) after the model is trained in the source domain. A good cross-domain bandit model can provide a head-start in this setting, therefore enjoying significantly lower cumulative regret in the target domain and saving substantial cost in collecting feedback in the high-cost target domain.

Fig. 1 shows the cumulative regrets of different methods on DIGIT, VisDA17 and S2RDA49, respectively. We show results averaged over five runs with different random seeds. Our DABand consistently

surpasses all baselines for all rounds in terms of both cumulative regrets and their increase rates, demonstrating DABand's potential to significantly reduce the cost in high-cost target domains.

**Ablation Study.** To verify the effectiveness of each term in our DABand's regret bound (Theorem 3.1, we report the accuracy of our proposed DABand after removing the **R**egression Error term and/or the **P**redicted Reward term (during training) in Table 4 for DIGIT, VisDA17 and S2RDA49. Results on all datasets show that removing either term will lead to performance drop. For example, in VisDA17, our full DABand achieves

Table 4: Results of the ablation study in terms of accuracy (higher is better). Note that the accuracy $ACC = 1 - \frac{1}{N}R_{\mathcal{T}}$, where $R_{\mathcal{T}}$ is the target regret. "R" and "P" is short for "Regression Error" and "Predicted Reward", respectively.

| Datasets | w/o R & P | w/o R | w/o P | DABand (Full) |
|----------|-----------|--------|--------|---------------|
| DIGIT    | 0.5676    | 0.5682 | 0.5768 | **0.6002**    |
| VisDA17  | 0.4088    | 0.4096 | 0.4304 | **0.4644**    |
| S2RDA49  | 0.3691    | 0.3694 | 0.3719 | **0.3923**    |

accuracy of $0.4642$; the accuracy drops to $0.4238$ after removing the Regression Error (R) term, and further drops to $0.4088$ after removing the Predicted Reward (P) term. These results verify the effectiveness of these two terms in our DABand.

## 6 CONCLUSION

In this paper, we introduce a novel domain adaptive neural contextual bandit algorithm, DABand, which adeptly combines effective exploration with representation alignment, utilizing unlabeled data from both source and target domains. Our theoretical analysis demonstrates that DABand can attain a sub-linear regret bound within the target domain. This marks the first regret analysis for domain adaptation in contextual bandit problems incorporating deep representation, shallow exploration, and adversarial alignment. We show that all these elements are instrumental in the domain adaptive bandit setting on real-world datasets. Moving forward, interesting future research could be to uncover more innovative techniques for aligning the source and target domains (mentioned in Appendix F), particularly within the constraints of: (1) bandit settings, as opposed to classification settings, and (2) the high-dimensional and dense nature of the target domain, contrasted with the sparse and simplistic nature of the source domain. It would also be interesting to explore the effect of improved uncertainty quantification (Wang & Wang, 2023), potentially via Bayesian deep learning (Wang & Yeung, 2016; 2020; Huang et al., 2019), on DABand's performance.

## ACKNOWLEDGMENTS

We sincerely appreciate the generous support from the reviewers/AC for the constructive comments to improve the paper. ZW and XH are partially sponsored by a subcontract of NSF grant 2229876, the A. Russell Chandler III Professorship at Georgia Institute of Technology, and NIH-sponsored Georgia Clinical & Translational Science Alliance. HW is partially supported by the Microsoft Research AI & Society Fellowship, NSF Grant IIS-2127918, NSF CAREER Award IIS-2340125, NIH Grant 1R01CA297832, and the Amazon Faculty Research Award. This research is also supported by NSF National Artificial Intelligence Research Resource (NAIRR) Pilot and the Frontera supercomputer, funded by the National Science Foundation (award NSF-OAC 1818253) and hosted at the Texas Advanced Computing Center (TACC) at The University of Texas at Austin. Finally, we extend our gratitude to the Center for AI Safety (CAIS) for providing the essential computing resources that made this work possible.

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

# A PROOFS

## A.1 PROOFS FOR LEMMAS

**Lemma A.1.** *For any hypotheses $h, h' \in \mathcal{H}$ and $a \in \mathcal{A}$ is selected by either $h$ or $h'$,*

$$|\epsilon_{\mathcal{S}}(h, h') - \epsilon_{\mathcal{T}}(h, h')| \leq \frac{N}{2} \widehat{d}_{\mathcal{H}\Delta\mathcal{H}}(\mathcal{S}, \mathcal{T}).$$

*Proof.* By definition of $\mathcal{H}\Delta\mathcal{H}$ (as defined in Def. 3.6), we have

$$
\begin{aligned}
\widehat{d}_{\mathcal{H}\Delta\mathcal{H}}(\mathcal{S}, \mathcal{T}) \quad &\overset{\text{Def. 3.6}}{=} \quad 2 \sup_{h,h' \in \mathcal{H}} |\mathbb{E}_{\mathbf{x} \sim \mathcal{S}}[|h(\mathbf{x}) - h'(\mathbf{x})|] - \mathbb{E}_{\mathbf{x} \sim \mathcal{T}}[|h(\mathbf{x}) - h'(\mathbf{x})|]| \\
&= \quad \frac{2}{N} \sup_{h,h' \in \mathcal{H}} \left| N \cdot \mathbb{E}_{\mathbf{x} \sim \mathcal{S}}[|h(\mathbf{x}) - h'(\mathbf{x})|] - N \cdot \mathbb{E}_{\mathbf{x} \sim \mathcal{T}}[|h(\mathbf{x}) - h'(\mathbf{x})|] \right| \\
&\overset{\text{Def. 3.3}}{=} \quad \frac{2}{N} \sup_{h,h' \in \mathcal{H}} \left| \epsilon_{\mathcal{S}}(h, h') - \epsilon_{\mathcal{T}}(h, h') \right| \\
&\geq \quad \frac{2}{N} \left| \epsilon_{\mathcal{S}}(h, h') - \epsilon_{\mathcal{T}}(h, h') \right|.
\end{aligned}
$$

$\square$

**Lemma A.2.** *Given three hypotheses $h_1, h_2, h_3 \in \mathcal{H}$, we have the following triangle inequality:*

$$\epsilon_{\mathcal{D}}(h_1, h_2) \leq \epsilon_{\mathcal{D}}(h_1, h_3) + \epsilon_{\mathcal{D}}(h_2, h_3). \tag{12}$$

*Proof.* The hypothesis difference

$$
\begin{aligned}
\epsilon_{\mathcal{D}}(h_1, h_2) \quad &= \quad \sum_{i=1}^{N} \left| h_1(x^{\mathcal{D}}) - h_2(x^{\mathcal{D}}) \right| \\
&\overset{\text{Def. 3.2}}{=} \quad \sum_{i=1}^{N} \left| h_1(x^{\mathcal{D}}) - h_2(x^{\mathcal{D}}) + h_3(x^{\mathcal{D}}) - h_3(x^{\mathcal{D}}) \right| \\
&\leq \quad \sum_{i=1}^{N} \left| h_1(x^{\mathcal{D}}) - h_3(x^{\mathcal{D}}) \right| + \sum_{i=1}^{N} \left| h_3(x^{\mathcal{D}}) - h_2(x^{\mathcal{D}}) \right| \\
&= \quad \epsilon_S(h_1, h_3) + \epsilon_S(h_2, h_3).
\end{aligned}
$$

$\square$

**Lemma A.3.** *Let $\mathcal{H}$ be a hypothesis space. Then for every $h \in \mathcal{H}$:*

$$\epsilon_{\mathcal{T}}(h) \leq \epsilon_{\mathcal{S}}(h) + \frac{N}{2} \widehat{d}_{\mathcal{H}\Delta\mathcal{H}}(\mathcal{S}, \mathcal{T}) + \psi, \tag{13}$$

*where $\psi$ is defined in Definition 3.7.*

*Proof.* By definition in Definition 3.3 in main paper, we start from $\epsilon_{\mathcal{T}}(h) = \epsilon_{\mathcal{T}}(h, f_{\mathcal{T}})$, then we have

$$
\begin{aligned}
\epsilon_{\mathcal{T}}(h) \quad &= \quad \epsilon_{\mathcal{T}}(f_{\mathcal{T}}, h) \\
&\overset{\text{Lm. A.2}}{\leq} \quad \epsilon_{\mathcal{T}}(f_{\mathcal{T}}, h^*) + \epsilon_{\mathcal{T}}(h, h^*) \\
&\leq \quad \epsilon_{\mathcal{T}}(h^*) + \epsilon_{\mathcal{S}}(h, h^*) + |\epsilon_{\mathcal{T}}(h, h^*) - \epsilon_{\mathcal{S}}(h, h^*)| \\
&\overset{\text{Lm. A.1}}{\leq} \quad \epsilon_{\mathcal{T}}(h^*) + \epsilon_{\mathcal{S}}(h, h^*) + \frac{N}{2} \widehat{d}_{\mathcal{H}\Delta\mathcal{H}}(\mathcal{S}, \mathcal{T}) \\
&\leq \quad \epsilon_{\mathcal{T}}(h^*) + \epsilon_{\mathcal{S}}(h) + \epsilon_{\mathcal{S}}(h^*) + \frac{N}{2} \widehat{d}_{\mathcal{H}\Delta\mathcal{H}}(\mathcal{S}, \mathcal{T}) \\
&= \quad \epsilon_{\mathcal{S}}(h) + \psi + \frac{N}{2} \widehat{d}_{\mathcal{H}\Delta\mathcal{H}}(\mathcal{S}, \mathcal{T}).
\end{aligned}
$$

$\square$

**Lemma A.4.** *Denoting the contexts from the target domain $\mathcal{T}$ as $\{\mathbf{x}_{t,a}^{\mathcal{T}}\}_{i\in[N],a\in[K]}$ and the associated ground-truth action as $\{a_i^*\}_{i=1}^N$, the estimated regret (i.e., using the trained model $\widehat{\theta}$ from the source domain) for the target domain is*

$$\sum_{i=1}^N \left( \left| \langle \widehat{\theta}, \widehat{\phi}(\mathbf{x}_{i,a_i^*}^{\mathcal{T}}) \rangle - \langle \widehat{\theta}, \widehat{\phi}(\mathbf{x}_{i,\widehat{a}_i}^{\mathcal{T}}) \rangle \right| \right) \leq \sum_{i=1}^N \left( \left| \langle \widehat{\theta}, \widehat{\phi}(\mathbf{x}_{i,\widehat{a}_i}^{\mathcal{T}}) \rangle \right| \right) + \sum_{i=1}^N \alpha \cdot \kappa_i^{\mathcal{T}},$$

*where $\kappa_i^{\mathcal{T}}$ is* $\max\left\{ 2\left\| \widehat{\phi}(\mathbf{x}_{i,\widehat{a}_i}^{\mathcal{T}}) \right\|_{A_{i-1}^{-1}}, \left\| \widehat{\phi}(\mathbf{x}_{i,a_i^*}^{\mathcal{T}}) \right\|_{A_{i-1}^{-1}} + \left\| \widehat{\phi}(\mathbf{x}_{i,\widehat{a}_i}^{\mathcal{T}}) \right\|_{A_{i-1}^{-1}} \right\}.$

*Proof.* By definition, we have

$$\sum_{i=1}^N \left( \left| \langle \widehat{\theta}, \widehat{\phi}(\mathbf{x}_{i,a_i^*}^{\mathcal{T}}) \rangle - \langle \widehat{\theta}, \widehat{\phi}(\mathbf{x}_{i,\widehat{a}_i}^{\mathcal{T}}) \rangle \right| \right)$$

$$= \sum_{i=1}^N \left( \left| \langle \widehat{\theta}, \widehat{\phi}(\mathbf{x}_{i,a_i^*}^{\mathcal{T}}) \rangle - \langle \widehat{\theta}, \widehat{\phi}(\mathbf{x}_{i,\widehat{a}_i}^{\mathcal{T}}) \rangle + \alpha \left\| \widehat{\phi}(\mathbf{x}_{i,\widehat{a}_i}^{\mathcal{T}}) \right\|_{A_{i-1}^{-1}} - \alpha \left\| \widehat{\phi}(\mathbf{x}_{i,\widehat{a}_i}^{\mathcal{T}}) \right\|_{A_{i-1}^{-1}} + \alpha \left\| \widehat{\phi}(\mathbf{x}_{i,a_i^*}^{\mathcal{T}}) \right\|_{A_{i-1}^{-1}} - \alpha \left\| \widehat{\phi}(\mathbf{x}_{i,a_i^*}^{\mathcal{T}}) \right\|_{A_{i-1}^{-1}} \right| \right)$$

$$= \sum_{i=1}^N \left( \left| \left( \langle \widehat{\theta}, \widehat{\phi}(\mathbf{x}_{i,a_i^*}^{\mathcal{T}}) \rangle + \alpha \left\| \widehat{\phi}(\mathbf{x}_{i,a_i^*}^{\mathcal{T}}) \right\|_{A_{i-1}^{-1}} \right) - \left( \langle \widehat{\theta}, \widehat{\phi}(\mathbf{x}_{i,\widehat{a}_i}^{\mathcal{T}}) \rangle + \alpha \left\| \widehat{\phi}(\mathbf{x}_{i,\widehat{a}_i}^{\mathcal{T}}) \right\|_{A_{i-1}^{-1}} \right) \right| \right)$$

$$+ \sum_{i=1}^N \alpha \left( \left| \left\| \widehat{\phi}(\mathbf{x}_{i,\widehat{a}_i}^{\mathcal{T}}) \right\|_{A_{i-1}^{-1}} - \left\| \widehat{\phi}(\mathbf{x}_{i,a_i^*}^{\mathcal{T}}) \right\|_{A_{i-1}^{-1}} \right| \right)$$

$$\leq \sum_{i=1}^N \left( \left| \left( \langle \widehat{\theta}, \widehat{\phi}(\mathbf{x}_{i,a_i^*}^{\mathcal{T}}) \rangle + \alpha \left\| \widehat{\phi}(\mathbf{x}_{i,a_i^*}^{\mathcal{T}}) \right\|_{A_{i-1}^{-1}} \right) - \left( \langle \widehat{\theta}, \widehat{\phi}(\mathbf{x}_{i,\widehat{a}_i}^{\mathcal{T}}) \rangle + \alpha \left\| \widehat{\phi}(\mathbf{x}_{i,\widehat{a}_i}^{\mathcal{T}}) \right\|_{A_{i-1}^{-1}} \right) \right| \right)$$

$$+ \sum_{i=1}^N \alpha \max\left\{ \left\| \widehat{\phi}(\mathbf{x}_{i,\widehat{a}_i}^{\mathcal{T}}) \right\|_{A_{i-1}^{-1}}, \left\| \widehat{\phi}(\mathbf{x}_{i,a_i^*}^{\mathcal{T}}) \right\|_{A_{i-1}^{-1}} \right\} \tag{14}$$

$$\leq \sum_{i=1}^N \left( \left| \left( \langle \widehat{\theta}, \widehat{\phi}(\mathbf{x}_{i,\widehat{a}_i}^{\mathcal{T}}) \rangle + \alpha \left\| \widehat{\phi}(\mathbf{x}_{i,\widehat{a}_i}^{\mathcal{T}}) \right\|_{A_{i-1}^{-1}} \right) \right| \right) + \sum_{i=1}^N \alpha \max\left\{ \left\| \widehat{\phi}(\mathbf{x}_{i,\widehat{a}_i}^{\mathcal{T}}) \right\|_{A_{i-1}^{-1}}, \left\| \widehat{\phi}(\mathbf{x}_{i,a_i^*}^{\mathcal{T}}) \right\|_{A_{i-1}^{-1}} \right\} \tag{15}$$

$$\leq \sum_{i=1}^N \left( \left| \langle \widehat{\theta}, \widehat{\phi}(\mathbf{x}_{i,\widehat{a}_i}^{\mathcal{T}}) \rangle \right| \right) + \sum_{i=1}^N \alpha \cdot \max\left\{ 2\left\| \widehat{\phi}(\mathbf{x}_{i,\widehat{a}_i}^{\mathcal{T}}) \right\|_{A_{i-1}^{-1}}, \left\| \widehat{\phi}(\mathbf{x}_{i,a_i^*}^{\mathcal{T}}) \right\|_{A_{i-1}^{-1}} + \left\| \widehat{\phi}(\mathbf{x}_{i,\widehat{a}_i}^{\mathcal{T}}) \right\|_{A_{i-1}^{-1}} \right\}$$

$$:= \sum_{i=1}^N \left( \left| \langle \widehat{\theta}, \widehat{\phi}(\mathbf{x}_{i,\widehat{a}_i}^{\mathcal{T}}) \rangle \right| \right) + \sum_{i=1}^N \alpha \cdot \kappa_i^{\mathcal{T}}, \tag{16}$$

where the inequality in Eqn. (14) is based on the fact that for any $a, b > 0$, $|a - b| \leq \max\{a, b\}$. The inequality in Eqn. (15) is derived from the definition of $\widehat{a}_i$, since $\widehat{a}_i$ is selected from the maximum value of $\left( \langle \widehat{\theta}, \widehat{\phi}(\mathbf{x}_{i,a}^{\mathcal{T}}) \rangle + \alpha \left\| \widehat{\phi}(\mathbf{x}_{i,a}^{\mathcal{T}}) \right\|_{A_{i-1}^{-1}} \right)$, therefore we guarantee that $\left( \langle \widehat{\theta}, \widehat{\phi}(\mathbf{x}_{i,\widehat{a}_i}^{\mathcal{T}}) \rangle + \alpha \left\| \widehat{\phi}(\mathbf{x}_{i,\widehat{a}_i}^{\mathcal{T}}) \right\|_{A_{i-1}^{-1}} \right) \geq \left( \langle \widehat{\theta}, \widehat{\phi}(\mathbf{x}_{i,a_i^*}^{\mathcal{T}}) \rangle + \alpha \left\| \widehat{\phi}(\mathbf{x}_{i,a_i^*}^{\mathcal{T}}) \right\|_{A_{i-1}^{-1}} \right)$. In addition, Eqn. (16) holds by definition as we define $\kappa_i^{\mathcal{T}}$ as $\max\left\{ 2\left\| \widehat{\phi}(\mathbf{x}_{i,\widehat{a}_i}^{\mathcal{T}}) \right\|_{A_{i-1}^{-1}}, \left\| \widehat{\phi}(\mathbf{x}_{i,a_i^*}^{\mathcal{T}}) \right\|_{A_{i-1}^{-1}} + \left\| \widehat{\phi}(\mathbf{x}_{i,\widehat{a}_i}^{\mathcal{T}}) \right\|_{A_{i-1}^{-1}} \right\}$. □

Next, we discuss the property of $\kappa_i^{\mathcal{T}}$ in the following lemma.

**Lemma A.5.** *For any arbitrary domain $\mathcal{D}$, we denote*

$\kappa_i^{\mathcal{D}} = \max\left\{ 2\left\| \widehat{\phi}(\mathbf{x}_{i,\widehat{a}_i}^{\mathcal{D}}) \right\|_{A_{i-1}^{-1}}, \left\| \widehat{\phi}(\mathbf{x}_{i,a_i^*}^{\mathcal{D}}) \right\|_{A_{i-1}^{-1}} + \left\| \widehat{\phi}(\mathbf{x}_{i,\widehat{a}_i}^{\mathcal{D}}) \right\|_{A_{i-1}^{-1}} \right\}.$ *Then, by our DABand algorithm, as the time step $i \to \infty$, we have $\kappa_i^{\mathcal{D}} \to 0$.*

*Proof.* By definition of $A$, its singular value increase as $i$ increase, which means that the singular values in its inverse matrix, $A^{-1}$, will decrease to 0. Therefore, for any new array $u$, $\|u\|_{A^{-1}} =$

$\sqrt{uA^{-1}u^T} \to 0$. Then when $i \to \infty$, we have

$$\lim_{i \to \infty} \kappa_i^{\mathcal{D}} = \max \left\{ 2\left\|\widehat{\phi}(\mathbf{x}_{i,\widehat{a}_i}^{\mathcal{D}})\right\|_{A_{i-1}^{-1}}, \left\|\widehat{\phi}(\mathbf{x}_{i,a_i^*}^{\mathcal{D}})\right\|_{A_{i-1}^{-1}} + \left\|\widehat{\phi}(\mathbf{x}_{i,\widehat{a}_i}^{\mathcal{D}})\right\|_{A_{i-1}^{-1}} \right\}$$
$$\approx 0.$$

$\square$

**Lemma A.6.** *Denoting the contexts from the source domain $\mathcal{S}$ as $\{\mathbf{x}_{t,a}^{\mathcal{S}}\}_{i \in [N], a \in [K]}$, the associated ground-truth action as $\{a_i^*\}_{i=1}^N$, we have the following inequality:*

$$\sum_{i=1}^N \left( \left| \langle \widehat{\theta}, \widehat{\phi}(\mathbf{x}_{i,a_i^*}^{\mathcal{S}}) \rangle - \langle \theta^*, \phi^*(\mathbf{x}_{i,\widehat{a}_i}^{\mathcal{S}}) \rangle \right| \right) \leq \epsilon_{\mathcal{S}}(h) + \sum_{i=1}^N \mathbb{1}[a_i^* \neq \widehat{a}_i] \left( \left( \left| \langle \widehat{\theta}, \widehat{\phi}(\mathbf{x}_{i,\widehat{a}_i}^{\mathcal{S}}) \rangle \right| \right) + \alpha \cdot \kappa_i^{\mathcal{S}} \right),$$

*where $\kappa_i^{\mathcal{S}}$ is defined in Lemma A.5.*

*Proof.* By definition, we have

$$\sum_{i=1}^N \left( \left| \langle \widehat{\theta}, \widehat{\phi}(\mathbf{x}_{i,a_i^*}^{\mathcal{S}}) \rangle - \langle \theta^*, \phi^*(\mathbf{x}_{i,\widehat{a}_i}^{\mathcal{S}}) \rangle \right| \right)$$

$$= \sum_{i=1}^N \left[ \mathbb{1}[a_i^* = \widehat{a}_i] \left( \left| \langle \widehat{\theta}, \widehat{\phi}(\mathbf{x}_{i,a_i^*}^{\mathcal{S}}) \rangle - \langle \theta^*, \phi^*(\mathbf{x}_{i,\widehat{a}_i}^{\mathcal{S}}) \rangle \right| \right) + \mathbb{1}[a_i^* \neq \widehat{a}_i] \left( \left| \langle \widehat{\theta}, \widehat{\phi}(\mathbf{x}_{i,a_i^*}^{\mathcal{S}}) \rangle - \langle \theta^*, \phi^*(\mathbf{x}_{i,\widehat{a}_i}^{\mathcal{S}}) \rangle \right| \right) \right]$$

$$\leq \sum_{i=1}^N \left[ \mathbb{1}[a_i^* = \widehat{a}_i] \left( \left| \langle \widehat{\theta}, \widehat{\phi}(\mathbf{x}_{i,\widehat{a}_i}^{\mathcal{S}}) \rangle - 1 \right| \right) + \mathbb{1}[a_i^* \neq \widehat{a}_i] \left( \left| \langle \widehat{\theta}, \widehat{\phi}(\mathbf{x}_{i,a_i^*}^{\mathcal{S}}) \rangle - 0 \right| \right) \right]$$

$$\leq \sum_{i=1}^N \left[ \mathbb{1}[a_i^* = \widehat{a}_i] \left( \left| \langle \widehat{\theta}, \widehat{\phi}(\mathbf{x}_{i,\widehat{a}_i}^{\mathcal{S}}) \rangle - 1 \right| \right) + \mathbb{1}[a_i^* \neq \widehat{a}_i] \left( \left| \langle \widehat{\theta}, \widehat{\phi}(\mathbf{x}_{i,a_i^*}^{\mathcal{S}}) \rangle + \langle \widehat{\theta}, \widehat{\phi}(\mathbf{x}_{i,\widehat{a}_i}^{\mathcal{S}}) \rangle - \langle \widehat{\theta}, \widehat{\phi}(\mathbf{x}_{i,\widehat{a}_i}^{\mathcal{S}}) \rangle \right| \right) \right]$$

$$\leq \sum_{i=1}^N \left[ \mathbb{1}[a_i^* = \widehat{a}_i] \left( \left| \langle \widehat{\theta}, \widehat{\phi}(\mathbf{x}_{i,\widehat{a}_i}^{\mathcal{S}}) \rangle - 1 \right| \right) + \mathbb{1}[a_i^* \neq \widehat{a}_i] \left( \left| \langle \widehat{\theta}, \widehat{\phi}(\mathbf{x}_{i,\widehat{a}_i}^{\mathcal{S}}) \rangle - 0 \right| \right) + \mathbb{1}[a_i^* \neq \widehat{a}_i] \left( \left| \langle \widehat{\theta}, \widehat{\phi}(\mathbf{x}_{i,a_i^*}^{\mathcal{S}}) \rangle - \langle \widehat{\theta}, \widehat{\phi}(\mathbf{x}_{i,\widehat{a}_i}^{\mathcal{S}}) \rangle \right| \right) \right]$$

$$\overset{\text{L.m. A.4}}{\leq} \sum_{i=1}^N \left( \left| \langle \theta_{\mathcal{S}}^*, \phi_{\mathcal{S}}^*(\mathbf{x}_{i,\widehat{a}_i}^{\mathcal{S}}) \rangle - \langle \widehat{\theta}, \widehat{\phi}(\mathbf{x}_{i,\widehat{a}_i}^{\mathcal{S}}) \rangle \right| \right) + \sum_{i=1}^N \mathbb{1}[a_i^* \neq \widehat{a}_i] \left( \left| \langle \widehat{\theta}, \widehat{\phi}(\mathbf{x}_{i,\widehat{a}_i}^{\mathcal{S}}) \rangle \right| + \alpha \cdot \kappa_i^{\mathcal{S}} \right)$$

$$\equiv \epsilon_{\mathcal{S}}(h) + \sum_{i=1}^N \mathbb{1}[a_i^* \neq \widehat{a}_i] \left( \left| \langle \widehat{\theta}, \widehat{\phi}(\mathbf{x}_{i,\widehat{a}_i}^{\mathcal{S}}) \rangle \right| + \alpha \cdot \kappa_i^{\mathcal{S}} \right),$$

$\square$

## A.2 PROOF FOR THE TARGET REGRET BOUND

**Theorem A.1 (Target Regret Bound).** *Denoting the contexts from the source domain $\mathcal{S}$ as $\{\mathbf{x}_{t,a}^{\mathcal{S}}\}_{i \in [N], a \in [K]}$, the associated ground-truth action as $\{a_i^*\}_{i=1}^N$, and the contexts from the target domain $\mathcal{T}$ as $\{\mathbf{x}_{t,a}^{\mathcal{T}}\}_{i \in [N], a \in [K]}$, the upper bound for our target regret $R_{\mathcal{T}}$ is*

$$R_{\mathcal{T}} \triangleq \sum_{i=1}^N \left( \left| \langle \theta_{\mathcal{T}}^*, \phi_{\mathcal{T}}^*(\mathbf{x}_{i,a_i^*}^{\mathcal{T}}) \rangle - \langle \theta_{\mathcal{T}}^*, \phi_{\mathcal{T}}^*(\mathbf{x}_{i,\widehat{a}_i}^{\mathcal{T}}) \rangle \right| \right)$$

$$\leq \underbrace{R_{\mathcal{S}}}_{\text{Source Regret}} + \underbrace{2 \cdot \epsilon_{\mathcal{S}}(h)}_{\text{Regression Error}} + \underbrace{N \cdot \widehat{d}_{\mathcal{H}\Delta\mathcal{H}}(\mathcal{S}, \mathcal{T})}_{\text{Data Divergence}} + \underbrace{\psi + C}_{\text{Constant}}$$

$$+ \underbrace{\sum_{i=1}^N \left( \left| \langle \widehat{\theta}, \widehat{\phi}(\mathbf{x}_{i,\widehat{a}_i}^{\mathcal{T}}) \rangle \right| \right) + \sum_{i=1}^N \mathbb{1}[a_i^* \neq \widehat{a}_i] \left( \left( \left| \langle \widehat{\theta}, \widehat{\phi}(\mathbf{x}_{i,\widehat{a}_i}^{\mathcal{S}}) \rangle \right| \right) \right)}_{\text{Predicted Rewards}},$$

*where $R_{\mathcal{S}}$, $\epsilon_{\mathcal{S}}(h)$, and $\widehat{d}_{\mathcal{H}\Delta\mathcal{H}}$ are the source regret, source-domain error, and $\mathcal{H}\Delta\mathcal{H}$ divergence defined in Definition 3.4, Definition 3.3, and Definition 3.6, respectively. $\psi$ is a constant independent to the problem and and $C$ is a constant which can be ignored.*

*Proof.* Please check the full proof in the next page.

By the definition of the Target Regret Bound, we have

$$R_{\mathcal{T}} = \sum_{i=1}^{N}\left(\left|\langle\theta^*,\phi^*(\mathbf{x}_{i,a_i^*}^{\mathcal{T}})\rangle - \langle\theta^*,\phi^*(\mathbf{x}_{i,\widehat{a}_i}^{\mathcal{T}})\rangle\right|\right)$$

$$= \sum_{i=1}^{N}\left(\left|\langle\theta^*,\phi^*(\mathbf{x}_{i,a_i^*}^{\mathcal{T}})\rangle - \langle\theta^*,\phi^*(\mathbf{x}_{i,\widehat{a}_i}^{\mathcal{T}})\rangle + \langle\widehat{\theta},\widehat{\phi}(\mathbf{x}_{i,\widehat{a}_i}^{\mathcal{T}})\rangle - \langle\widehat{\theta},\widehat{\phi}(\mathbf{x}_{i,\widehat{a}_i}^{\mathcal{T}})\rangle\right|\right)$$

$$\le \sum_{i=1}^{N}\left(\left|\langle\theta^*,\phi^*(\mathbf{x}_{i,\widehat{a}_i}^{\mathcal{T}})\rangle - \langle\widehat{\theta},\widehat{\phi}(\mathbf{x}_{i,\widehat{a}_i}^{\mathcal{T}})\rangle\right|\right) + \sum_{i=1}^{N}\left(\left|\langle\theta^*,\phi^*(\mathbf{x}_{i,a_i^*}^{\mathcal{T}})\rangle - \langle\widehat{\theta},\widehat{\phi}(\mathbf{x}_{i,\widehat{a}_i}^{\mathcal{T}})\rangle\right|\right)$$

$$\le \epsilon_{\mathcal{T}}(h) + \sum_{i=1}^{N}\left(\left|\langle\theta^*,\phi^*(\mathbf{x}_{i,a_i^*}^{\mathcal{T}})\rangle - \langle\widehat{\theta},\widehat{\phi}(\mathbf{x}_{i,\widehat{a}_i}^{\mathcal{T}})\rangle + \langle\widehat{\theta},\widehat{\phi}(\mathbf{x}_{i,a_i^*}^{\mathcal{T}})\rangle - \langle\widehat{\theta},\widehat{\phi}(\mathbf{x}_{i,a_i^*}^{\mathcal{T}})\rangle\right|\right)$$

$$\le \epsilon_{\mathcal{T}}(h) + \sum_{i=1}^{N}\left(\left|\langle\widehat{\theta},\widehat{\phi}(\mathbf{x}_{i,a_i^*}^{\mathcal{T}})\rangle - \langle\widehat{\theta},\widehat{\phi}(\mathbf{x}_{i,\widehat{a}_i}^{\mathcal{T}})\rangle\right|\right) + \sum_{i=1}^{N}\left(\left|\langle\theta^*,\phi^*(\mathbf{x}_{i,a_i^*}^{\mathcal{T}})\rangle - \langle\widehat{\theta},\widehat{\phi}(\mathbf{x}_{i,a_i^*}^{\mathcal{T}})\rangle\right|\right)$$

$$\le \epsilon_{\mathcal{T}}(h) + \sum_{i=1}^{N}\left(\left|\langle\widehat{\theta},\widehat{\phi}(\mathbf{x}_{i,a_i^*}^{\mathcal{T}})\rangle - \langle\widehat{\theta},\widehat{\phi}(\mathbf{x}_{i,\widehat{a}_i}^{\mathcal{T}})\rangle\right|\right) + \sum_{i=1}^{N}\left(\left|\langle\theta^*,\phi^*(\mathbf{x}_{i,a_i^*}^{\mathcal{T}})\rangle - \langle\widehat{\theta},\widehat{\phi}(\mathbf{x}_{i,a_i^*}^{\mathcal{T}})\rangle\right|\right)$$
$$+ \sum_{i=1}^{N}\left(\left|\langle\theta^*,\phi^*(\mathbf{x}_{i,a_i^*}^{\mathcal{S}})\rangle - \langle\widehat{\theta},\widehat{\phi}(\mathbf{x}_{i,a_i^*}^{\mathcal{S}})\rangle\right|\right) - \sum_{i=1}^{N}\left(\left|\langle\theta^*,\phi^*(\mathbf{x}_{i,a_i^*}^{\mathcal{S}})\rangle - \langle\widehat{\theta},\widehat{\phi}(\mathbf{x}_{i,a_i^*}^{\mathcal{S}})\rangle\right|\right)$$

$$\le \epsilon_{\mathcal{T}}(h) + \sum_{i=1}^{N}\left(\left|\langle\widehat{\theta},\widehat{\phi}(\mathbf{x}_{i,a_i^*}^{\mathcal{T}})\rangle - \langle\widehat{\theta},\widehat{\phi}(\mathbf{x}_{i,\widehat{a}_i}^{\mathcal{T}})\rangle\right|\right) + \sum_{i=1}^{N}\left(\left|\langle\theta^*,\phi^*(\mathbf{x}_{i,a_i^*}^{\mathcal{S}})\rangle - \langle\widehat{\theta},\widehat{\phi}(\mathbf{x}_{i,a_i^*}^{\mathcal{S}})\rangle\right|\right)$$
$$+ \left|\sum_{i=1}^{N}\left(\left|\langle\theta^*,\phi^*(\mathbf{x}_{i,a_i^*}^{\mathcal{T}})\rangle - \langle\widehat{\theta},\widehat{\phi}(\mathbf{x}_{i,a_i^*}^{\mathcal{T}})\rangle\right|\right) - \sum_{i=1}^{N}\left(\left|\langle\theta^*,\phi^*(\mathbf{x}_{i,a_i^*}^{\mathcal{S}})\rangle - \langle\widehat{\theta},\widehat{\phi}(\mathbf{x}_{i,a_i^*}^{\mathcal{S}})\rangle\right|\right)\right|$$

$$\le \epsilon_{\mathcal{T}}(h) + \frac{N}{2}\widehat{d}_{\mathcal{H}\Delta\mathcal{H}}(\mathcal{S},\mathcal{T}) + \sum_{i=1}^{N}\left(\left|\langle\widehat{\theta},\widehat{\phi}(\mathbf{x}_{i,a_i^*}^{\mathcal{T}})\rangle - \langle\widehat{\theta},\widehat{\phi}(\mathbf{x}_{i,\widehat{a}_i}^{\mathcal{T}})\rangle\right|\right)$$
$$+ \sum_{i=1}^{N}\left(\left|\langle\theta^*,\phi^*(\mathbf{x}_{i,a_i^*}^{\mathcal{S}})\rangle - \langle\widehat{\theta},\widehat{\phi}(\mathbf{x}_{i,a_i^*}^{\mathcal{S}})\rangle + \langle\theta^*,\phi^*(\mathbf{x}_{i,\widehat{a}_i}^{\mathcal{S}})\rangle - \langle\theta^*,\phi^*(\mathbf{x}_{i,\widehat{a}_i}^{\mathcal{S}})\rangle\right|\right)$$

$$\overset{\text{L.m.. A.3}}{\le} \epsilon_{\mathcal{S}}(h) + \psi + \frac{N}{2}\widehat{d}_{\mathcal{H}\Delta\mathcal{H}}(\mathcal{S},\mathcal{T}) + \frac{N}{2}\widehat{d}_{\mathcal{H}\Delta\mathcal{H}}(\mathcal{S},\mathcal{T}) + \sum_{i=1}^{N}\left(\left|\langle\widehat{\theta},\widehat{\phi}(\mathbf{x}_{i,a_i^*}^{\mathcal{T}})\rangle - \langle\widehat{\theta},\widehat{\phi}(\mathbf{x}_{i,\widehat{a}_i}^{\mathcal{T}})\rangle\right|\right)$$
$$+ \sum_{i=1}^{N}\left(\left|\langle\theta^*,\phi^*(\mathbf{x}_{i,a_i^*}^{\mathcal{S}})\rangle - \langle\theta^*,\phi^*(\mathbf{x}_{i,\widehat{a}_i}^{\mathcal{S}})\rangle\right|\right) + \sum_{i=1}^{N}\left(\left|\langle\widehat{\theta},\widehat{\phi}(\mathbf{x}_{i,a_i^*}^{\mathcal{S}})\rangle - \langle\theta^*,\phi^*(\mathbf{x}_{i,\widehat{a}_i}^{\mathcal{S}})\rangle\right|\right)$$

$$\overset{\text{L.m. A.4}}{\le} \epsilon_{\mathcal{S}}(h) + \psi + N\widehat{d}_{\mathcal{H}\Delta\mathcal{H}}(\mathcal{S},\mathcal{T}) + \sum_{i=1}^{N}\left(\left|\langle\widehat{\theta},\widehat{\phi}(\mathbf{x}_{i,\widehat{a}_i}^{\mathcal{T}})\rangle\right|\right) + \sum_{i=1}^{N}\alpha\cdot\kappa_i^{\mathcal{T}}$$
$$+ R_{\mathcal{S}} + \sum_{i=1}^{N}\left(\left|\langle\widehat{\theta},\widehat{\phi}(\mathbf{x}_{i,a_i^*}^{\mathcal{S}})\rangle - \langle\theta^*,\phi^*(\mathbf{x}_{i,\widehat{a}_i}^{\mathcal{S}})\rangle\right|\right)$$

$$\overset{\text{L.m. A.6}}{\le} \epsilon_{\mathcal{S}}(h) + \psi + N\widehat{d}_{\mathcal{H}\Delta\mathcal{H}}(\mathcal{S},\mathcal{T}) + \sum_{i=1}^{N}\left(\left|\langle\widehat{\theta},\widehat{\phi}(\mathbf{x}_{i,\widehat{a}_i}^{\mathcal{T}})\rangle\right|\right) + \sum_{i=1}^{N}\alpha\cdot\kappa_i^{\mathcal{T}}$$
$$+ R_{\mathcal{S}} + \epsilon_{\mathcal{S}}(h) + \sum_{i=1}^{N}\mathbb{1}[a_i^*\ne\widehat{a}_i]\left(\left|\langle\widehat{\theta},\widehat{\phi}(\mathbf{x}_{i,\widehat{a}_i}^{\mathcal{S}})\rangle\right| + \alpha\cdot\kappa_i^{\mathcal{S}}\right)$$

$$\overset{\text{L.m. A.5}}{=} \underbrace{R_{\mathcal{S}}}_{Source\ Regret} + \underbrace{2\cdot\epsilon_{\mathcal{S}}(h)}_{Regression\ Error} + \underbrace{N\cdot\widehat{d}_{\mathcal{H}\Delta\mathcal{H}}(\mathcal{S},\mathcal{T})}_{Data\ Divergence} + \underbrace{\psi + C}_{Constant}$$
$$+ \underbrace{\sum_{i=1}^{N}\left(\left|\langle\widehat{\theta},\widehat{\phi}(\mathbf{x}_{i,\widehat{a}_i}^{\mathcal{T}})\rangle\right|\right) + \mathbb{1}[a_i^*\ne\widehat{a}_i]\left(\sum_{i=1}^{N}\left(\left|\langle\widehat{\theta},\widehat{\phi}(\mathbf{x}_{i,\widehat{a}_i}^{\mathcal{S}})\rangle\right|\right)\right)}_{Predicted\ Rewards},$$

where $C = \sum_{i=1}^{N} \alpha \cdot \kappa_i^{\mathcal{T}} + \mathbb{1}[a_i^* \neq \widehat{a}_i] \left( \sum_{i=1}^{N} \alpha \cdot \kappa_i^{\mathcal{S}} \right)$ is a small number which can be ignored as $i \to \infty$ (i.e., see Lemma A.5). □

## B   DATASETS

To evaluate the effectiveness of our DABand, for each dataset, we treat the "easy" domain as the low-cost source domain and the more challenging one as the high-cost target domain. This allows us to demonstrate the efficacy of our method by adapting from a simpler to a more complex domain within a contextual bandit setting.

*DIGIT.* Our DIGIT dataset consists of MNIST and MNIST-M. MNIST is a gray-scale hand-written digit dataset, while MNIST-M (Ganin et al., 2016a) features color digits. In this paper, MNIST serves as our source domain, and MNIST-M as our target domain, offering a more challenging adaptation path. The MNIST digits are gray-scale and uniform in size, aspect ratio, and intensity range, in stark contrast to the colorful and varied digits of MNIST-M. Therefore, adapting from MNIST to MNIST-M presents a greater challenge than adapting from MNIST-M to MNIST, where the target domain is less complex.

*VisDA17.* The VisDA-2017 image classification challenge (Peng et al., 2017) addresses a domain adaptation problem across 12 classes, involving three distinct datasets. The training set consists of 3D rendering images, whereas the validation and test sets feature real images from the COCO (Lin et al., 2014) and YouTube Bounding Boxes (Real et al., 2017) datasets, respectively. Ground truth labels were provided only for the training and validation sets. Scores for the test set were calculated by a server operated by the competition organizers. For our purposes, we use the training set as the source domain and the validation set as the target domain.

*S2RDA49.* The S2RDA49 (Synthetic-to-Real) is a new benchmark dataset (Tang & Jia, 2023) constructed in 2023. This dataset contains 49 classes. The source domain (i.e., the synthetic domain) is synthesized by rendering 3D models from ShapeNet (Chang et al., 2015). The used 3D models are in the same label space as the target/real domain, and each class has 12K rendered RGB images. The target domain (i.e., the real domain) of S2RDA49 contains 60535 images from 49 classes, collected from the ImageNet validation set (Deng et al., 2009), ObjectNet (Barbu et al., 2019), VisDA2017 validation set (Peng et al., 2017), and the web. In this paper, we select the only 10 class that matches the VisDA17 dataset.

## C   BASELINES

We compare our DABand with both classic and state-of-the-art contextual bandit algorithms. We start with **LinUCB** (Li et al., 2010), a typical baseline for linear contextual bandit problems. To enable comprehensive comparisons, we extend our evaluation to include LinUCB augmented with pre-processed features using principle component analysis (PCA), referred to as **LinUCB-P**. In LinUCB-P, we first apply PCA to fit on training data (without labels/feedback) in both source and target domain. Then, with transformed, lower-dimensional context data, LinUCB is performed to see whether this pre-alignment procedure can transfer the knowledge to the target domain. Another pivotal baseline in our study is Neural-LinUCB (Xu et al., 2020) (**NLinUCB**), which utilizes several layers of fully connected neural networks to dynamically process the original features at the beginning of every iteration. Similar to LinUCB-P, we introduce a NLinUCB variant that incorporates PCA, i.e., **NLinUCB-P**. Note that domain adaptation baselines are **not applicable** to our setting. Specifically, domain adaptation methods only work in offline settings, and assume complete observability of labels in the source domain. In contrast, contextual bandit is an online setting where the oracle is revealed **only when correctly predicted**. Therefore domain adaptation methods are not applicable to our online bandit settings.

## D   IMPLEMENTATION DETAILS

In this section, we provide detailed insights into the implementation of our approach, applied to two distinct datasets: DIGIT and VisDA17. We use Pytorch to implement our method, and all experiments are run on servers with NVIDA A5000 GPUs.

**DIGIT.** Within the DIGIT dataset framework, we use the MNIST dataset as the source domain and MNIST-M as the target domain. To ensure compatibility between the datasets, we standardize the channel size of images in the source domain ($c_{\mathcal{S}} = 1$) to align with that in the target domain ($c_{\mathcal{T}} = 3$). Each image undergoes normalization and is resized to $28 \times 28$ pixels with 3 channels to accommodate the format requirements of both domains. Then, an encoder is utilized to diminish the data's dimensionality to a more manageable latent space. Following this reduction, the data is processed through two fully connected neural network layers, ending in the final latent space necessary for loss computation as delineated in main paper. For the optimal hyperparameters, we set the learning rate to $1 \times e^{-5}$, with $\lambda$ is chosen from $\{1.0, 5.0, 10.0, 15.0, 20.0\}$ and kept the same for all experiments. Additionally, we set the exploration rate $\alpha$ to 0.05.

**VisDA17.** We use the VisDA17 dataset's training set as the source domain, with the validation set functioning as the target domain. We adhere to preprocessing steps established by (Prabhu et al., 2021) to ensure uniformity across domains: Each image is normalized and resized to $224 \times 224$ pixels with 3 channels, matching the requisite specifications for both source and target domains. For hyperparameters, the learning rate of $1 \times e^{-5}$ is applied, with $\lambda$ is chosen from $\{1.0, 5.0, 10.0, 15.0, 20.0\}$ and then kept the same for all experiments. We set the exploration rate $\alpha$ to 0.05.

**S2RDA49.** We selects 10 classes from the original 49 classes since it matches the target domain samples in VisDA17 (Peng et al., 2017). We adhere to preprocessing steps established by (Prabhu et al., 2021) to ensure uniformity across domains: Each image is normalized and resized to $224 \times 224$ pixels with 3 channels. For hyperparameters, the learning rate of $1 \times e^{-3}$ is applied, with $\lambda$ chosen from $\{1.0, 5.0, 10.0, 15.0, 20.0\}$ and then kept the same for all experiments. We set the exploration rate $\alpha$ to 0.01.

Table 5: Results of the ablation studies in terms of accuracy (higher is better). Note that the accuracy $ACC = 1 - \frac{1}{N}R_{\mathcal{T}}$, where $R_{\mathcal{T}}$ is the target regret. "R", "P" and "D" are short for "Regression Error", "Predicted Reward" and "Data Divergence" (i.e., adversarial loss and the discriminator), respectively.

| Datasets | w/o R&P&D | w/o R&P | w/o R&D | w/o P&D | w/o R | w/o P | w/o D | DABand (Full) |
|---|---|---|---|---|---|---|---|---|
| DIGIT | 0.3816±0.04 | 0.5676±0.02 | 0.3793±0.02 | 0.3544±0.01 | 0.5682±0.01 | 0.5768±0.01 | 0.3649±0.02 | **0.6002**±0.02 |
| VisDA17 | 0.1001±0.02 | 0.4088±0.03 | 0.1010±0.02 | 0.0936±0.01 | 0.4096±0.01 | 0.4304±0.01 | 0.1098±0.01 | **0.4644**±0.03 |
| S2RDA49 | 0.1108±0.02 | 0.3691±0.02 | 0.0918±0.01 | 0.1032±0.01 | 0.3694±0.03 | 0.3719±0.02 | 0.1121±0.02 | **0.3923**±0.03 |

# E   FULL ABLATION STUDIES

The full ablation study is shown in Table 5. Furthermore, we ran the corresponding hypothesis tests, and the p values are in the range of $(3.201 \times 10^{-21}, 1.504 \times 10^{-2})$, much lower than the threshold of 0.05 and therefore verifying the significance of DABand's performance improvement.

# F   LIMITATION

This work contains several limitations. Specifically, we highlight below:

**Intuitive Perspective.** The bandit algorithm indeed enjoys interpretability (Wang et al., 2024b;a) and the accuracy of its closed-form updates (Li et al., 2010), but this is true only because it typically employs a linear model.

Unfortunately, linear models do not work very well for real-world data, which is often high-dimensional. For example, the empirical results for LinUCB and LinUCB-P in Table 1 and Table 2 show that such linear contextual bandit algorithms significantly underperform state-of-the-art neural bandit algorithms, which use back-propagation (similar results are shown in (Zhou et al., 2019; Xu et al., 2020)). In summary deep learning models with back-propagation is necessary due to the following reasons:

- **High-Dimensional Data.** To enhance performance in real-world high-dimensional data (e.g., images), the integration of deep learning (and back-propagation) is necessary, though this may sacrifice a portion of the algorithm's explanatory power.

- **Covariate Shift and Aligning Source and Target Domains in the Latent Space.** In our settings where there is covariance shift between the source and target domains, a deep (nonlinear) encoder is required to transform the original context into a latent space where source-domain encodings and target-domain encoders can align. This also necessitates deep learning models with back-propagation.

Indeed, there is a trade-off between interpretability and performance. This would certainly be an interesting future direction, but it is out of the scope of this paper.

**Theoretical Perspective.** Our Theorem 3.1 is sharp, as all the inequalities are based on lemmas in the paper and the Cauchy inequality. Identifying the criteria under which the target regret bound reaches equality as well as how one can achieve it in practice would be interesting future work.

**Empirical Perspective.** The performance of DABand largely relies on the alignment quality between the source and target domains. Therefore, for two domains that cannot be aligned (for example, most domain adaptation tasks are predefined, and the data for both domains is pre-processed and cleaned, not original real-world data), it is challenging to evaluate how DABand can still transfer knowledge across different domains. Furthermore, it is still unknown whether, if we increase the domain shift in a dataset from another domain, our DABand can still perform well. These issues are beyond the scope of this paper, but they represent interesting areas for future work.

## G  DISCUSSIONS

### G.1  SUBLINEARITY FOR THE TARGET REGRET BOUND

To see the data divergence term is sub-linear, note that our target regret bound in Eqn. (5) can be divided into two parts:

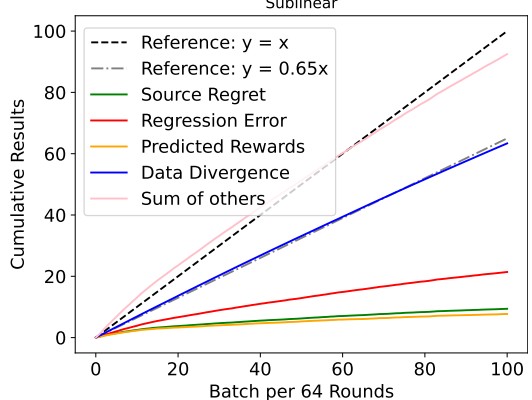

Figure 2: Sub-linearity for each term in Eqn. (5).

- **Total Source Regret**: In Neural-LinUCB (NLinUCB) (Xu et al., 2020), this is shown to be $O(\sqrt{N})$, which is sub-linear.

- **Sum of Other Terms (Including the Data Divergence)**: These terms can be directly optimized to convergence and minimized to a small value using SGD variants such as Adam. As shown in Corollary 4.2 of the Adam paper (Diederik, 2014), the Adam optimizer enjoys an average regret $R(N)/N$ of $O(\frac{1}{\sqrt{N}})$ (here $N$ is equivalent to $T$ in the Adam paper), which leads to a sub-linear total regret $O(\sqrt{N})$. Since all other terms are directly optimized using Adam, they also enjoy a sub-linear regret.

Therefore, our target regret bound is also sub-linear. Moreover:

- **Key Difference between the Source Regret and Other Terms.** Note that in our target regret bound in Eqn. (5), the source regret term $R_{\mathcal{S}}$ increases monotonically with respect to $N$ and **cannot** be directly minimized. In contrast, all other terms, e.g., the data divergence term $N \cdot \widehat{d}_{\mathcal{H}\Delta\mathcal{H}}(\mathcal{S}, \mathcal{T})$ **can** be directly minimized since all related data is training data and is already known. This is why we minimize the corresponding loss terms like Eqn. (7), Eqn. (8) and Eqn. (9) during training.

- The data divergence term can be minimized to a small value as $N \to \infty$. For example, if the source and target domains are perfectly aligned after our training, the data divergence becomes 0. Therefore, the cumulative sum of the data divergence term over all rounds will be sub-linear.

## G.2 Difference between Bandit and Classification Settings

In the classification setting, given a sample $\mathbf{x}$, a model predicts its label $\widehat{y}$. Since its ground-truth label is known, one can directly apply the cross-entropy loss to perform back-propagation and update our model. In contrast, in bandit settings, for each sample $\mathbf{x}$, our model predicts an action $\widehat{a}$ by optimizing the estimated rewards. One then submits this predicted action to the environment and receives feedback that only indicates whether the predicted action $\widehat{a}$ is the optimal action $a^*$ or not. If not, we are informed that our prediction is incorrect, yet we do not receive information on what the correct (optimal) action should be. Moreover, during training, we only train on 300 episodes for DIGIT and VisDA17 and 100 episodes for S2RDA49. Each episode contains 64 samples. This implies that compared with those DA methods which train on multiple epochs, in our settings, all of the models only see a sample **once**. Such complexity makes the bandit setting much more challenging.

**Why Simultaneously Training Source and Target Domains is Needed and Helpful.** It is also noteworthy that our DABand is an **end-to-end** model that enables **two-way feedback between source and target domains**:

- **Source to Target:** Collecting **source**-domain rewards helps DABand to learn a better encoder (which transforms raw contexts into encodings in the latent space) because the reward signals help the encoder extract **more relevant encodings** (embeddings) from **target**-domain contexts, thereby **reducing the target-domain regret**.

- **Target to Source:** After aligning source-domain and **target-domain** encodings in the shared latent space (i.e., minimizing the data divergence term in Eqn. (5)), the information from the **target**-domain context can then help the **source** domain to explore arms that are **more relevant to the target domain**. Ultimately, this also helps reduce the target-domain regret even without collecting target-domain rewards.

## G.3 Importance of Bandits and DABand

Bandit algorithms are designed to navigate environments where feedback is sparse, costly, and indirect. By efficiently learning from limited feedback – identifying not just when a prediction is wrong but adapting without explicit guidance on the right choice – bandit algorithms offer a strategic advantage in dynamically evolving settings. Our DABand exemplifies this by leveraging a low-cost source domain to improve performance in a high-cost target domain, thereby reducing cumulative regret (improving accuracy) while minimizing operational costs. DABand not only reduces the expense associated with acquiring and labeling vast datasets but also capitalizes on the intrinsic adaptability of bandit algorithms to learn and optimize in complex, uncertain environments.

## G.4 Novelties Restatement

**Theoretical Novelty.** In general, DABand is the **first** work to perform contextual bandit in a domain adaptation setting, i.e., adapt from a source domain with feedback to a target domain without feedback. Moreover, our theoretical analysis presents two major technical challenges/novelties below:

- **Generalization of $\mathcal{H}\Delta\mathcal{H}$ Distance to Regression.** In the original multi-domain generalization bound (Ben-David et al., 2010), the $\mathcal{H}\Delta\mathcal{H}$ divergence between source and target domains is derived for classification models. However, contextual bandit is essentially a reward regression problem, and therefore necessitates generalizing the $\mathcal{H}\Delta\mathcal{H}$ distance from the classification case to the regression case. This presents a significant technical challenge, and leads to a series of modifications in the proof.

- **Decomposing the Target Regret into the Source Regret with Other Terms.** The subsequent major challenge our DABand addresses involves decomposing the regret bound for the target domain (i.e., the target regret) into a source regret term and other terms to upper-bound the target regret. This is necessary because we do not have access to feedback/reward from the target domain (we only have access to feedback/reward in the source domain). This process requires a nuanced understanding of the interplay between source and target domain dynamics within our model's framework.

**Other Technical Novelties.** Moreover, our contribution goes beyond the theoretical analysis itself. Specifically,

- We identify the problem of contextual bandits across domains and propose domain-adaptive contextual bandits (DABand) as the first general method to explore a high-cost target domain while only collecting feedback from a low-cost source domain.
- Our theoretical analysis shows that our method can achieve a sub-linear regret bound in the target domain.
- Our empirical results on real-world datasets show our DABand significantly improves performance over the state-of-the-art contextual bandit methods when adapting across domains.

### G.5 Discussion on Zero-shot Target Regret Bound

#### G.5.1 Significance of Theorem 3.1

Note that Theorem 3.1 is **nontrivial**. While it does resemble the generalization bound in domain adaptation, there are key differences. As mentioned in Observation (3) in Sec. 3.6, our target regret bound includes two additional crucial terms not found in domain adaptation. Specifically:

- **Regression Error in the Source Domain.** $\sum_{i=1}^{N} \left( \left| \langle \theta_{\mathcal{S}}^*, \phi_{\mathcal{S}}^*(x_{i,\widehat{a}_i}^{\mathcal{S}}) \rangle - \langle \widehat{\theta}, \widehat{\phi}(x_{i,\widehat{a}_i}^{\mathcal{S}}) \rangle \right| \right)$, which defines the difference between the true reward from selecting action $\widehat{a}_i$ and the estimated reward for this action.
- **Predicted Reward.** $\sum_{i=1}^{N} \left( \left| \langle \widehat{\theta}, \widehat{\phi}(\mathbf{x}_{i,\widehat{a}_i}^{\mathcal{T}}) \rangle \right| \right) + \sum_{i=1}^{N} \mathbb{1}[a_i^* \neq \widehat{a}_i] \left( \left| \langle \widehat{\theta}, \widehat{\phi}(\mathbf{x}_{i,\widehat{a}_i}^{\mathcal{S}}) \rangle \right| \right)$, which serves as a regularization term to regularize the model to avoid overestimating rewards.

The results of the ablation study in Table 4 in Sec. 5 highlight the significance of these two terms. Please also refer to "Technical Novelty" of Sec. G.4 above for discussion on key novelty/challenges in deriving Theorem 3.1.

#### G.5.2 Testing Phase rather than Traditional Bandit Learning Settings

Our regret bound in Theorem 3.1 is a zero-shot regret bound for the target domain (with empirical results in the **Zero-Shot Target Regret** paragraph of Sec. 5), which corresponds to a testing phase.

However, we would like to clarify that extending this bound to handle continued training in the target domain (corresponding to the **Continued Training in Target Domains and Cumulative Regret** paragraph of Sec. 5) is straightforward. At a high level, we can have

$$R_{\mathcal{T}}^{total} = R_{\mathcal{T}}^{zero-shot} + R_{\mathcal{T}}^{continued},$$

with the first term handled by our DABand's Theorem 3.1 and the second term handled by typical single-domain contextual bandit.

#### G.5.3 Scale of Source Regret and Predicted Rewards

Similar to Neural-LinUCB (NLinUCB), in our DABand, the true reward is restricted to the range of $[0, 1]$ (as we mentioned in Sec. 3.1); therefore it will not make the source regret unbounded.

Furthermore, our DABand algorithm tries to minimize the source regret while aligning the source and target domains. The minimization of the source regret is theoretically guaranteed, as discussed in recent work such as Neural-LinUCB (NLinUCB) (Xu et al., 2020).

### G.6 Are the Comparisons Fair?

**Leveraging Target-Domain Contexts.** A lot of our baselines **do leverage** the contexts of the target domain. For example, our baseline LinUCB-P starts by performing PCA jointly on both source-domain and target-domain contexts and then perform LinUCB. Therefore LinUCB-P does leverage the contexts of the target domain. Similarly, NLinUCB-P starts by performing PCA jointly on both

source-domain and target-domain context and then perform NLinUCB. It therefore also leverages target-domain contexts.

**Seeing Target-Domain Rewards.** Note that in our domain adaptive bandit settings:

- During the **zero-shot** phase, target rewards are **not** visible for all methods (include both baselines and our DABand); it is therefore fair comparison.
- During the **continued-training** phase, all methods (include both baselines and our DABand) start to see the target reward and update their parameters; it is therefore also fair comparison.

### G.7 WHY MINIMIZE THE PREDICTED REWARD ON THE SOURCE DOMAIN?

While the predicted-reward term is naturally derived from our theoretical analysis, we do find interesting insights when examining this term and its relation to our model. Specifically:

- **Indirect Regularization on Bandit Parameters $\widehat{\theta}$ and the Encoder $\widehat{\phi}(\cdot)$.** One can see this term as an L1 regularization term. It does not directly regularize the bandit parameter $\widehat{\theta}$; however, minimizing the L1 norm of the predicted reward (i.e., a $K$-dimensional vector for a $K$-arm bandit) does **indirectly** regularize the bandit parameter $\widehat{\theta}$ and the encoder $\widehat{\phi}(\cdot)$, thereby preventing the L1 norm of the predicted reward from getting to large.
- **Smaller Predicted Rewards for Smaller Variance and Better Stability.** Furthermore, this regularization can help avoid predicting high rewards for all arms in the bandit.
  - Note that for the bandit algorithm to achieve low regret, predicting large rewards are not necessary. This is because one uses the **argmax** operation (i.e., Line 6 in Alg. 1) to select the best arm; an arm $k$ with a small predicted reward can still be selected as long as all other arms have even smaller predicted rewards.
  - Too higher predicted rewards are not desirable because they increase the model's sensitivity, leading to higher variance and subsequently increasing the generalization error (i.e., the target regret's bound).

### G.8 CLARIFICATION OF DABAND'S CONTRIBUTIONS

Our DABand is the first general method to explore a target domain while only collecting feedback from the source domain, regardless of linear or nonlinear assumptions. No prior methods have explored this setting of exploring a target domain while only collecting feedback from the source domain.

Therefore, DABand's contribution is two-fold: (1) DABand is the first general method to explore a target domain while only collecting feedback from the source domain; (2) DABand is also the first general method in this domain-adaptive bandit setting that works even under general nonlinear assumptions.

### G.9 IMPORTANCE OF THE DISCRIMINATOR.

Our bandit setting aims to explore a target domain while only collecting feedback from the source domain. **All reward feedback in the target domain is unknown**, making this approach very **challenging**. Therefore, using a discriminator to align representations for both domains is necessary. If we ignore the discriminator, the method will not work in our settings. This is also evidenced by the whole results for ablation studies in Table 5.

### G.10 HOW THE TERMS IN THE REGRET DECAY AS N INCREASES.

To see how the terms in the regret decay:

- **Decay in the Source Regret.** Most of the decay occurs in the first term (i.e., Source Regret $R\_\mathcal{S}$), which has been discussed in the NLinUCB paper (Xu et al., 2020). Specifically the **cumulative** source regret is $O(\sqrt{N})$. Then, dividing both sides of Eqn. (5) by $N$, we will get the **average** source regret term with $O(\sqrt{N}/N)$, which does decay with $N$.

- **Decay in the Data Divergence Term.** Decay in the Data Divergence Term. The data divergence term, $\hat{d}_{\mathcal{H}\Delta\mathcal{H}}(\mathcal{S}, \mathcal{T})$, can actually be further decomposed into
  - an empirical term that estimates the divergence using $N$ source-domain contexts and $N$ target-domain contexts, and
  - a term related to $N$ with complexity $\mathcal{O}(\log(2N)/N)$ (Ben-David et al., 2010).

  Therefore, the second term with $\mathcal{O}(\log(2N)/N)$ does also decay as $N$ increases.

### G.11 UNDERSTANDING THE BOUND IF THE SOURCE AND TARGET DOMAINS ARE EQUIVALENT

Even if the input distributions of the two domains match, it does not imply that the relationship between the input context $\mathbf{x}$ and the predicted reward is the same for them, since the data divergence term is only responsible for aligning the distribution of $\mathbf{x}$ (input); it is irrelevant to the output and rewards. This is why other terms are needed to characterize the difference in the relationship between the context $\mathbf{x}$ and the reward in the source and target domains. The data divergence term alone is not sufficient.

### G.12 CLARIFICATION ON THE PROBLEM SETTING.

**Simultaneous Observation.** In our setting, source-domain contexts and target-domain contexts are simultaneously observed, with only the reward of the source domain being observed.

**Target-Domain Contexts Available Even Before Running Alg. 1.** Note that in practice, target-domain contexts are usually **available** even before Algorithm 1 starts and **before observing any reward from the source domain**. (This makes it possible to train our DABand by simultaneously using source-domain and target-domain contexts.)

For example, in the case of testing drug reactions on mice (source domain) and humans (target domain), usually one already has the human subjects' genomics, demographic, and other data as target-domain contexts, even before testing the drug on mice (i.e., the source domain) and collecting rewards.

This is because these human genomics/demographic data are easy to collect at a low cost; in contrast, testing the drug on humans (i.e., the target domain) and collecting rewards involves extremely high costs, due to the risks of fatality, side effects, and the enormous costs of conducting clinical trials.

## H POTENTIAL IMPACT OF DABAND

Our DABand has many potential real-world applications, especially when obtaining responses is costly. For instance, in testing new drugs, we can construct responses from mice for new drug A, and then use DABand to obtain the zero-shot hypothetical regret bound for responses in humans. In another scenario, we can collect data (responses) on humans for another published, similar drug B, and then transfer knowledge by aligning the divergence between drug A and drug B. If both regrets reveal acceptable performance, we might not need excessive costs for back-and-forth testing, which significantly speeds up the process and reduces costs.

