# OpenReview forum: "Towards Domain Adaptive Neural Contextual Bandits"
_ICLR.cc/2025/Conference — ICLR 2025 Poster_

### Official Review · Reviewer_Mn7w · 2024-11-01

**Soundness:** 3
**Presentation:** 2
**Contribution:** 3
**Rating:** 6
**Confidence:** 3

**Summary:**

This paper studies the domain transfer in (neural) contextual bandits. The authors provide a decomposition of the domain transfer in contextual bandits, and proposed an algorithm leveraging the adversarial training in domain-transfer. Theoretical results are presented understanding this decomposition and empirical evaluation are performed to validate the performance of the algorithm

**Strengths:**

- The authors provide a theoretical justification on the decomposition for the domain-transfer issue for contextual bandits.
- The authors provide empirical evaluation demonstrating the performance of the algorithm.

**Weaknesses:**

- (Minor) Definition 3.6 is hard to understand for me. In particular, are $\mathbf x$ and $x$ referring to the same thing? Are the authors implicitly assumes $h \in \mathcal H$ and $x \in \mathbb R^n$ in this case? It would be helpful if the authors could rephrase the mathematical formulation on this.
- In Theorem 3.1 needs further justification. For example, I wonder why the sublinear in $R_S$ will lead to the sublinear result in $R_T$. I checked the proof for Theorem 3.1 in appendix but found nowhere explicitly discuss this part. In addition, why the data-divergence term is sublinear?
- The adversarial training paradigm adapted in controlling the data divergence term in the proposed algorithm requires better justification: why the hypothesis class here is the same with the hypothesis class used in contextual bandits?
- It lacks some discussions on the performance analysis in linear contextual bandits or discussions in related literautre. The authors might want to provide more discussion on the theoretical insight of this.

- (Minor) Section 4.1 title: eq (16) -> eq (5)

**Questions:**

- Besides the presentation in Theorem 3.1, are there any regret scale for linear contextual bandits to help me better understand the performance of the algorithm?

---

> ### Author Response · Authors · 2024-11-20
> **[1/2] Thank you for the constructive comments**
>
> Thank you for your constructive comments and for acknowledging that we have provided ``"theoretical justification"`` for ``"understanding this decomposition"`` and that ``"empirical evaluation are performed to validate the performance of the algorithm"``. Below, we address your questions one by one in detail. We have also **included all discussions below in our revision** (with the changed part marked in blue).
>
> **Q1: "(Minor) Definition 3.6 is hard to understand for me. In particular, are x and x referring to the same thing? Are the authors implicitly assumes and in this case? It would be helpful if the authors could rephrase the mathematical formulation on this."**
>
> We are sorry for the confusion. Yes, they are referring to the same thing. We have fixed this typo in the revision accordingly (marked in blue).
>
> **Q2: "In Theorem 3.1 needs further justification. For example, I wonder why the sublinear in R_S will lead to the sublinear result in R_T. I checked the proof for Theorem 3.1 in appendix but found nowhere explicitly discuss this part. In addition, why the data-divergence term is sublinear?"**
>
> This is a good question.
>
> **Theoretical Analysis on Sublinearity.** To see the target regret is sub-linear, note that our target regret bound, i.e.,
>
> $\mathit{R}\_{\mathcal{T}}  \leq \mathit{R}\_{\mathcal{S}} + 2 \cdot \epsilon\_{\mathcal{S}} (h)+ N \cdot \hat{d}\_{\mathcal{H}\Delta\mathcal{H}} (\mathcal{S}, \mathcal{T}) + \psi + C + \sum\_{i=1}^{N} \left( \left| \langle \hat{\theta}, \hat{\phi} (\mathbf{x}\_{i, \hat{a}\_{i}}^{\mathcal{T}}) \rangle \right| \right) + \sum\_{i=1}^{N} \mathbb{1}{[a\_{i}^{*} \neq \hat{a}\_{i}]} \left( \left| \langle \hat{\theta}, \hat{\phi} (\mathbf{x}\_{i, \hat{a}\_{i}}^{\mathcal{S}})  \rangle \right| \right)$,
>
> can be divided into two parts:
> + **Total Source Regret**: In NeuralLinUCB [1], this is shown to be $O(\sqrt{N})$, which is sublinear.
> + **Sum of Other Terms (Including the Data Divergence)**: These terms can be *directly optimized* to convergence and minimized to a small value using SGD variants such as Adam. As shown in Corollary 4.2 of the Adam paper [2], the Adam optimizer enjoys an average regret $R(N)/N$ of $O(\frac{1}{\sqrt{N}})$ (here $N$ is equivalent to $T$ in the Adam paper), which leads to a sub-linear total regret $O(\sqrt{N})$. Since all other terms are directly optimized using Adam, they also enjoy a sub-linear regret.
> Therefore, our target regret bound is also sublinear.
>
> To clarify any potential confusion, we have updated Algorithm 1 in the paper to explicitly indicate that the Adam optimizer is used to train both the encoder and the discriminator.
>
> **Key Difference between the Source Regret and Other Terms.** Note that in our target regret bound above, the source regret term $\mathit{R}\_{\mathcal{S}}$ increases monotonically with respect to $N$ and *cannot* be directly minimized. In contrast, all other terms, e.g., the data divergence term $N \cdot \hat{d}\_{\mathcal{H}\Delta\mathcal{H}} (\mathcal{S}, \mathcal{T})$ *can* be directly minimized since all related data is training data and is already known. This is why we minimize the corresponding loss terms, e.g., Eqn. (7), Eqn. (8), and Eqn. (9) in the paper during training.
>
> In response to your specific question, the data divergence term can be minimized to a small value as $N \to \infty$. For example, if the source and target domains are perfectly aligned after our training, the data divergence becomes $0$. Therefore, the cumulative sum of the data divergence term over all rounds will be sublinear.
>
> **Empirical Results on Sublinearity.** Besides the theoretical insights above, we have also empirically demonstrated that each term is sublinear. We have included a plot along with the above discussion in **Figure 2 of Appendix G.1** in our revised paper. The figures shows that each term is sub-linear with respect to $N$.

---

> ### Author Response · Authors · 2024-11-20
> **[2/2] Thank you for the constructive comments**
>
> **Q3: "The adversarial training paradigm adapted in controlling the data divergence term in the proposed algorithm requires better justification: why the hypothesis class here is the same with the hypothesis class used in contextual bandits?"**
>
> Thank you for mentioning this. In the original multi-domain generalization bound [3] for adversarial training paradigms, the hypothesis class is derived for *classification* models, i.e., $\mathbb{R}^d \to \{ 0, 1 \}$.
>
> In contrast, contextual bandit is essentially a reward *regression* problem, i.e., $\mathbb{R}^d \to [0,1] $. Therefore it is necessary to generalize the hypothesis class from the classification case to the regression case. This presents a significant technical challenge, leads to a series of modifications in our proof, and is actually one of our DABand's key contributions.
>
> In Definition 3.6 of the paper, we discussed such a generalization from classification to regression. This generalization is also critical when we prove our Theorem 3.1 on the target regret bound, specifically for the data divergence term in Eqn. (5).
>
> With our generalization, the hypothesis class of the adapted adversarial training paradigm in our DABand is then *the same* as that of contextual bandits, i.e., they are both essentially a regression model $\mathbb{R}^d \to [0,1] $.
>
> **Q4: "It lacks some discussions on the performance analysis in linear contextual bandits or discussions in related literautre. The authors might want to provide more discussion on the theoretical insight of this."**
>
> This is a good suggestion.
>
> Our target regret bound (Eqn. (5) of the paper) consists of 5 terms, i.e., source regret, regression error, data divergence, constant, and predicted reward.
>
> After applying linear contextual bandits on the source domain, the source regret enjoys a sub-linear rate. However, in the linear model, the encoder $\hat{\phi}(x) = x$ is an identity function and therefore fail to align source-domain and target-domain contexts, leading to an unbounded, large data divergence term in Eqn. (5) of the paper. This is the main reason why linear contextual bandits, such as LinUCB, perform poorly in the cross-domain contextual bandit setting.
>
> Note that even contextual bandits using a nonlinear encoder, e.g., NeuralLinUCB, fail in this case because their encoders $\hat{\phi}(x)$ are not learned to align source-domain and target-domain contexts; they therefore will still lead to an unbounded, large data divergence term in Eqn. (5) of the paper, and subsequently poor performance in the cross-domain contextual bandit setting.
>
> We have included the discussion above in our revision (Page 9 in the main paper) as suggested. We will also be very happy to add and discuss any other related literature you might suggest.
>
> **Q5: "(Minor) Section 4.1 title: eq (16) -> eq (5)"**
>
> We are sorry for the confusion. We have fixed this in the revision.
>
> [1] Xu et al. Neural contextual bandits with deep representation and shallow exploration. ICLR 2022
>
> [2] Kingma et al. Adam: A Method for Stochastic Optimization. ICLR 2015
>
> [3] Ben-David et al. A theory of learning from different domains. Machine Learning 2010.

---

> > ### Author Response · Authors · 2024-11-24
> > **Your Feedback Would Be Appreciated**
> >
> > Dear Reviewer Mn7w,
> >
> > Thank you once again for your valuable comments. Your suggestions on clarifying hypothesis class in problem setting and baseline performance analysis were very helpful. We are eager to know if our responses have adequately addressed your concerns.
> >
> > Due to the limited time for discussion, we look forward to receiving your feedback and hope for the opportunity to respond to any further questions you may have.
> >
> > Yours Sincerely,
> >
> > Authors of DABand

---

> > > ### Author Response · Authors · 2024-11-29
> > > **Your Feedback Would Be Appreciated**
> > >
> > > Dear Reviewer Mn7w,
> > >
> > > Due to the limited time for discussion, we look forward to receiving your feedback and hope for the opportunity to respond to any further questions you may have.
> > >
> > > Thank you again for your encouraing and valuable comments.
> > >
> > > Yours Sincerely,
> > >
> > > Authors of DABand

---

> > > > ### Comment · Reviewer_Mn7w · 2024-12-01
> > > >
> > > > Thank the authors for their feedback. It addressed most of my concerns so I will keep my original positive rating. I would suggest the authors to incorporate the discussions in the revision and discuss more on the context bandit tasks.

---

> > > > > ### Author Response · Authors · 2024-12-01
> > > > > **Thank you**
> > > > >
> > > > > Dear Reviewer Mn7w,
> > > > >
> > > > > Thank you once again for your encouraging and valuable feedback! We are glad that our response has addressed your concerns. We will be sure to include more discussions on the contextual bandit tasks in the final revision as you suggested.
> > > > >
> > > > > Best regards,
> > > > >
> > > > > DABand Authors

---

### Official Review · Reviewer_MPVh · 2024-11-04

**Soundness:** 3
**Presentation:** 3
**Contribution:** 3
**Rating:** 6
**Confidence:** 3

**Summary:**

The paper introduces an algorithm for domain adaptation in contextual bandits. This method aims to adapt a neural contextual bandit algorithm from a low-cost source domain to a high-cost target domain, to address the challenge of distribution shifts between the two domains. The theoretical analysis conducted demonstrates that DABand achieves a sub-linear regret bound in the target domain, which is interesting. Empirical results are presented using real-world datasets, which show that DABand outperforms existing contextual bandit methods. The proposed work has broader implications, including potential insights into reinforcement learning with human feedback algorithms.

**Strengths:**

(1) The proposed problem setting of contextual bandits in a domain adaptation scenario is interesting and challenging.

(2) The method utilizes unlabeled data from both source and target domains for effective representation learning and alignment across different domains.

(3) The algorithm is capable of attaining a sub-linear regret bound in the target domain by solving an online network lasso problem with time-dependent regularization.

**Weaknesses:**

The method leverages unlabeled data from both source and target domains to learn robust representations and aligns them effectively across different domains, enabling efficient domain adaptation. While the proposed algorithm builds upon the NeuralLinUCB framework, it introduces an adaptation in the loss function specifically tailored for updating the neural network. This loss function integrates insights from classic domain adaptation techniques, and thereby has some similarity with existing methods.

**Questions:**

See weakness.

---

> ### Author Response · Authors · 2024-11-20
> **Thank you for the constructive comments**
>
> Thank you for your constructive comments and insightful questions. We are glad that you found our problem setting ``"interesting and challenging"``, our theoretical results ``"interesting"``, our method ``"outperforms existing contextual bandit methods."`` and ``"has broader implications"``.
> Below, we address your questions one by one in detail. We have also **included all discussions below in our revision** (with the changed part marked in blue).
>
> **Q1: "... This loss function integrates insights from classic domain adaptation techniques, and thereby has some similarity with existing methods."**
>
> Thank you for your question. We would like to highlight our novelty of this work compared with **existing methods** (more details in Appendix G.4).
>
> **Theoretical Novelty.** In general, DABand is the **first** work to perform contextual bandit in a domain adaptation setting, i.e., adapt from a source domain with feedback to a target domain without feedback. Moreover, our theoretical analysis presents two major technical challenges/novelties below:
>
> + **Generalization of $H\Delta H$ Distance to Regression.** In the original multi-domain generalization bound [1] for domain adaptation, the $H\Delta H$ divergence between source and target domains is derived for *classification* models. However, contextual bandit is essentially a reward *regression* problem, and therefore necessitates generalizing the $H\Delta H$ distance from the classification case to the regression case. This presents a significant technical challenge, and leads to a series of modifications in our proof.
>
> + **Decomposing the Target Regret into the Source Regret with Other Terms.** The subsequent major challenge our DABand addresses involves decomposing the regret bound for the target domain (i.e., the target regret) into a source regret term and other terms to upper-bound the target regret. This is necessary because we do not have access to feedback/reward from the target domain (we only have access to feedback/reward in the source domain). This process requires a nuanced understanding of the interplay between source and target domain dynamics within our model's framework.
>
> **Other Technical Novelties.** We would also like to bring to the reviewer's attention that our contribution goes beyond the theoretical analysis itself. Specifically,
> + We identify the problem of contextual bandits across domains and propose domain-adaptive contextual bandits (DABand) as the first general method to explore a high-cost target domain while only collecting feedback from a low-cost source domain.
> + Our theoretical analysis shows that our method can achieve a sub-linear regret bound in the target domain.
> + Our empirical results on real-world datasets show our DABand significantly improves performance over the state-of-the-art contextual bandit methods when adapting across domains.
>
> More details are in **Appendix G.4**, and we have highlighted this section in the main paper as suggested, in case other readers find it helpful.
>
> [1] Ben-David et al. A theory of learning from different domains. Machine Learning 2010.

---

> > ### Author Response · Authors · 2024-11-24
> > **Your Feedback Would Be Appreciated**
> >
> > Dear Reviewer MPVh,
> >
> > Thank you once again for your valuable comments. Your suggestions on clarifying the difference between our method and some similar existing methods were very helpful. We are eager to know if our responses have adequately addressed your concerns.
> >
> > Due to the limited time for discussion, we look forward to receiving your feedback and hope for the opportunity to respond to any further questions you may have.
> >
> > Yours Sincerely,
> >
> > Authors of DABand

---

> > > ### Author Response · Authors · 2024-11-29
> > > **Your Feedback Would Be Appreciated**
> > >
> > > Dear Reviewer MPVh,
> > >
> > > Due to the limited time for discussion, we look forward to receiving your feedback and hope for the opportunity to respond to any further questions you may have.
> > >
> > > Thank you again for your encouraing and valuable comments.
> > >
> > > Yours Sincerely,
> > >
> > > Authors of DABand

---

### Official Review · Reviewer_2Moo · 2024-11-13

**Soundness:** 3
**Presentation:** 3
**Contribution:** 3
**Rating:** 8
**Confidence:** 3

**Summary:**

The manuscript introduces a domain adaption algorithm for contextual bandit. It further proves that the proposed method achieves a sub-linear regret bound. Empirical experiments conducted on real-world data shows the the proposed method outperforms existing state-of-the-art contextual bandit methods for cross domains.

**Strengths:**

1. The design of the new algorithm originates from an observation that leveraging data across domains leads to sub-linear regrets, which makes the whole method simple, yet elegant.

2. Theoretical proof has been provided to support the performance of the method.

3. The algorithm is extensively tested on three datasets.

**Weaknesses:**

It would be helpful to elaborate on why the data divergence term is sub-linear in the proof.

**Questions:**

See weakness.

---

> ### Author Response · Authors · 2024-11-20
> **Thank you for the constructive and encouraging comments**
>
> Thank you for your constructive comments and insightful questions. We are glad that you found our method ``"simple, yet elegant"``, that our theoretical proof ``"has been provided to support the performance of the method"``, and that our algorithm ``"extensively tested"``. Below, we address your questions in detail. We have also **included all discussions below in our revision** (with the changed part marked in blue).
>
> **Q1: "It would be helpful to elaborate on why the data divergence term is sub-linear in the proof."**
>
> This is a good question.
>
> **Theoretical Analysis on Sublinearity.** To see the data divergence term is sub-linear, note that our target regret bound, i.e.,
>
> $\mathit{R}\_{\mathcal{T}}  \leq \mathit{R}\_{\mathcal{S}} + 2 \cdot \epsilon\_{\mathcal{S}} (h)+ N \cdot \hat{d}\_{\mathcal{H}\Delta\mathcal{H}} (\mathcal{S}, \mathcal{T}) + \psi + C + \sum\_{i=1}^{N} \left( \left| \langle \hat{\theta}, \hat{\phi} (\mathbf{x}\_{i, \hat{a}\_{i}}^{\mathcal{T}}) \rangle \right| \right) + \sum\_{i=1}^{N} \mathbb{1}{[a\_{i}^{*} \neq \hat{a}\_{i}]} \left( \left| \langle \hat{\theta}, \hat{\phi} (\mathbf{x}\_{i, \hat{a}\_{i}}^{\mathcal{S}})  \rangle \right| \right)$,
>
> can be divided into two parts:
> + **Total Source Regret**: In NeuralLinUCB [1], this is shown to be $O(\sqrt{N})$, which is sublinear.
> + **Sum of Other Terms (Including the Data Divergence)**: These terms can be *directly optimized* to convergence and minimized to a small value using SGD variants such as Adam. As shown in Corollary 4.2 of the Adam paper [2], the Adam optimizer enjoys an average regret $R(N)/N$ of $O(\frac{1}{\sqrt{N}})$ (here $N$ is equivalent to $T$ in the Adam paper), which leads to a sub-linear total regret $O(\sqrt{N})$. Since all other terms are directly optimized using Adam, they also enjoy a sub-linear regret.
> Therefore, our target regret bound is also sublinear.
>
> To clarify any potential confusion, we have updated Algorithm 1 in the paper to explicitly indicate that the Adam optimizer is used to train both the encoder and the discriminator.
>
> **Key Difference between the Source Regret and Other Terms.** Note that in our target regret bound above, the source regret term $\mathit{R}\_{\mathcal{S}}$ increases monotonically with respect to $N$ and *cannot* be directly minimized. In contrast, all other terms, e.g., the data divergence term $N \cdot \hat{d}\_{\mathcal{H}\Delta\mathcal{H}} (\mathcal{S}, \mathcal{T})$ *can* be directly minimized since all related data is training data and is already known. This is why we minimize the corresponding loss terms, e.g., Eqn. (7), Eqn. (8), and Eqn. (9) in the paper during training.
>
>
> In response to your specific question, the data divergence term can be minimized to a small value as $N \to \infty$. For example, if the source and target domains are perfectly aligned after our training, the data divergence becomes $0$. Therefore, the cumulative sum of the data divergence term over all rounds will be sublinear.
>
>
> [1] Xu et al. Neural contextual bandits with deep representation and shallow exploration. ICLR 2022
>
> [2] Kingma et al. Adam: A Method for Stochastic Optimization. ICLR 2015

---

> > ### Author Response · Authors · 2024-11-24
> > **Your Feedback Would Be Appreciated**
> >
> > Dear Reviewer 2Moo,
> >
> > Thank you once again for your valuable comments. Your suggestions on clarifying why the data divergence term is sub-linear in the proof were very helpful. We are eager to know if our responses have adequately addressed your concerns.
> >
> > Due to the limited time for discussion, we look forward to receiving your feedback and hope for the opportunity to respond to any further questions you may have.
> >
> > Yours Sincerely,
> >
> > Authors of DABand

---

> > > ### Author Response · Authors · 2024-11-29
> > > **Your Feedback Would Be Appreciated**
> > >
> > > Dear Reviewer 2Moo,
> > >
> > > Due to the limited time for discussion, we look forward to receiving your feedback and hope for the opportunity to respond to any further questions you may have.
> > >
> > > Thank you again for your encouraing and valuable comments.
> > >
> > > Yours Sincerely,
> > >
> > > Authors of DABand

---

> > > > ### Comment · Reviewer_MPVh · 2024-12-02
> > > >
> > > > Thanks for the authors' rebuttal. I would like to keep my score.

---

> > > > > ### Author Response · Authors · 2024-12-03
> > > > > **Thank you**
> > > > >
> > > > > Dear Reviewer MPVh,
> > > > >
> > > > > Thank you once again for your encouraging and valuable feedback! We are glad that our response has addressed your concerns. We will be sure to incorporate your comments and our associated response in the final revision.
> > > > >
> > > > > Best regards,
> > > > >
> > > > > DABand Authors

---

### Author Response · Authors · 2024-11-20
**Global Response**

We thank all reviewers for their valuable comments.

We are glad that they found our problem setting ``"interesting and challenging"`` (MPVh), our method ``"simple, yet elegant"``/``"outperforms existing contextual bandit methods"``/``"has broader implications"`` (2Moo, MPVh), and our algorithm ``"extensively tested"``/``"provide empirical evaluation"`` (2Moo, Mn7w). They also acknowledged that ``"theoretical justification"``/``"theoretical analysis"`` ``"has been provided"``, is ``"interesting"``, and ``"support the performance of the method"`` (2Moo, MPVh, Mn7w).

Below we address the reviewers' questions one by one in detail. We have cited all related references and **included all discussions/results below in our revision** (with the changed part marked in blue).

---

### Meta-Review · Area_Chair_URqG · 2024-12-22

**Metareview:**

The reviewers generally appreciate the novel algorithm for domain adaptation in bandits. I only have one more thing to add. Domain adaptation ultimately is a form of distributional shift. Another approach to deal with is through distributional robustness (see Distributionally Robust Batch Contextual Bandits, Si et al). The authors should discuss the merits and drawbacks of the current work with respect to this approach. But overall a good paper.

**Additional Comments On Reviewer Discussion:**

NA

---

### Decision · Program_Chairs · 2025-01-22

Accept (Poster)